# Spatiotemporal distribution of seasonal snow water equivalent in High-Mountain Asia from an 18-year Landsat-MODIS era snow reanalysis dataset

Yufei Liu[1], Yiwen Fang[1] and Steven A. Margulis[1]

[1]Department of Civil and Environmental Engineering, University of California, Los Angeles, CA, USA

*Correspondence to*: Steven A. Margulis (margulis@seas.ucla.edu)

**Abstract**: Seasonal snowpack is an essential component in the hydrological cycle and plays a significant role in supplying water resources to downstream users. Yet the snow water equivalent (SWE) in seasonal snowpacks, and its space-time variation, remains highly uncertain, especially over mountainous areas with complex terrain and sparse observations, such as in High-Mountain Asia (HMA). In this work, we assessed the spatiotemporal distribution of seasonal SWE, obtained from a new 18-year HMA Snow Reanalysis (HMASR) dataset, as part of the recent NASA High-Mountain Asia Team (HiMAT) effort. A Bayesian snow reanalysis scheme previously developed to assimilate satellite derived fractional snow-covered area (fSCA) products from Landsat and MODIS platforms has been applied to develop the HMASR dataset (at a spatial resolution of 16 arc-second (~500 m) and daily temporal resolution) over the joint Landsat-MODIS period covering Water Years (WYs) 2000-2017.

Based on the results, the HMA-wide total SWE volume is found to be around 163 km$^3$ on average and ranges from 114 km$^3$ (WY2001) to 227 km$^3$ (WY2005) when assessed over 18 WYs. The most abundant snowpacks are found in the northwestern basins (e.g. Indus, Syr Darya and Amu Darya) that are mainly affected by the westerlies, accounting for around 66% of total seasonal SWE volume. Seasonal snowpack in HMA is depicted by snow accumulating through October to March and April, typically peaking around April and depleting in July-October, with variations across basins and WYs. When examining the elevational distribution over the HMA domain, seasonal SWE volume peaks at mid-elevations (around 3500 m), with over 50% of the volume stored above 3500 m. Above-average amounts of precipitation causes significant overall increase in SWE volumes across all elevations; while an increase in air temperature (~ 1.5 K) from cooler to normal conditions leads to an redistribution in snow storage from lower elevations to mid elevations.

This work brings new insight into understanding the climatology and variability of seasonal snowpack over HMA, with the regional snow reanalysis constrained by remote sensing data, providing a new reference dataset for future studies of seasonal snow and how it contributes to the water cycle and climate over the HMA region.

# 1 Introduction

The High-Mountain Asia (HMA) region consists of the major mountain ranges and headwaters of the largest rivers in Asia. It features extremely high elevation, complex topography and significant glacier and snow cover. In HMA, glacier melt and snowmelt are vital to the hydrological cycle and water supply, as they feed the major regional rivers with over one billion people living downstream (Barnett et al., 2005; Bookhagen and Burbank, 2010; Immerzeel et al., 2010; Immerzeel and Bierkens, 2012; Lutz et al., 2014; Armstrong et al., 2019; Scott et al., 2019; Immerzeel et al., 2020).

Even though both seasonal snow and glaciers are crucial to hydrology and water availability, seasonal snow has arguably received less attention than glaciers in the HMA region. Many studies have addressed the status and changes in glaciers over HMA (e.g. Bolch et al., 2012; Kääb et al., 2012; Sorg et al., 2012; Yao et al., 2012; Lutz et al., 2014; Bolch et al., 2019; Rounce et al., 2020; Shean et al, 2020). For seasonal snow, previous studies have examined the snow extent (e.g. Dahe et al., 2006; Pu et al., 2007; Immerzeel et al., 2009; Tahir et al., 2011; Basang et al., 2017; Wang et al., 2017; Notarnicola 2020), or snow mass and snow depth (e.g. Dahe et al., 2006; Che et al., 2008; Terzago et al., 2014; Dai et al., 2017; Stigter et al., 2017; Smith and Bookhagen, 2018. 2020; Ahmad et al., 2019; Kirkham et al., 2019; Xue et al., 2019; Bair et al., 2018, 2020, 2021). In the current literature involving seasonal snow, most of the studies have focused on snow covered area (or extent, which is readily available from satellite-borne remote sensing) instead of snow mass, or have been applied at relatively localized scales (e.g. individual small to moderate sized basins) or coarse scales (e.g. above 1 km) over larger scales. The seasonal snow water storage and its spatiotemporal distribution across HMA are highly uncertain, primarily due to the lack of in situ observations and fine-scale (e.g. < 1 km) snow water equivalent (SWE) datasets over this large domain (Takala et al., 2011; Kirkham et al., 2019). In fact, accurately estimating SWE at such scales remains a great challenge worldwide, and it is even more difficult in mountainous regions due to the terrain complexity (Lettenmaier et al., 2015; Dozier et al., 2016; Bormann et al., 2018).

In situ measurements are usually expensive and difficult to install and maintain in HMA and are mostly located in low-lying valleys, thus resulting in a sparse and potentially nonrepresentative network (Winiger et al., 2005; Palazzi et al., 2013; Dozier et al., 2016; Kirkham et al., 2019). In recent decades, satellite observations can provide large-scale estimates of some snowpack properties. However, most of these measured properties, such as snow-covered area (SCA) based on visible and near infrared bands (e.g. Dozier, 1989, Hall et al., 2002, Painter et al., 2009), are only indirectly related to snow mass. While SWE and snow depth can be directly estimated from passive microwave sensors (using retrieval algorithms based on the brightness temperature; e.g. Chang et al., 1987), these estimates are at coarse spatial resolution (e.g. 25 km), and are generally negatively biased in deep snowpacks (Takala et al., 2011; Dozier et al, 2016). Recent applications of C-band synthetic aperture radar (SAR) techniques show promise for snow depth retrieval (Lievens et al., 2019) but are available only over recent years and do not directly provide SWE.

Global atmospheric reanalysis products provide another approach to large-scale SWE estimates as by-products of their land surface schemes. Examples include the Global Land Data Assimilation System (GLDAS, Rodell et al., 2004), Modern-Era

Retrospective analysis for Research and Applications (MERRA, Rienecker et al., 2011; MERRA-2, Gelaro et al., 2017), European Centre for Medium-Range Weather Forecasts (ECMWF) Re-Analysis products (ERA-Interim, Dee et al., 2011; ERA5, Hersbach et al., 2020), High Asia Refined analysis (HAR, Maussion et al., 2011; 2014), Japanese 55-year Reanalysis (JRA-55; Kobayashi et al., 2015), and others. SWE estimates in these datasets are found to be generally consistent in their interannual and seasonal variations, but can differ significantly in their magnitudes when evaluated over different regions (Mudryk et al., 2015; Wrzesien et al., 2019), where the uncertainties come from different land surface models and meteorological inputs (Mudryk et al., 2015; Mortimer et al., 2020; Kim et al., 2021). In addition, most reanalysis datasets are not specifically designed for SWE estimation, and only a few of them (e.g. ERA5 and JRA55) assimilate snow observations (including in-situ and remote sensing) in HMA. Bian et al. (2019) found many reanalysis datasets overestimate SWE compared to ground observations in the Tibetan Plateau, although part of the differences may come from inconsistent spatial resolution and elevations between in situ and gridded datasets. The performance of these large-scale reanalysis datasets over the full HMA domain has not been fully assessed due to the sparse and uneven in situ station network.

Recent works have contributed to the development of SWE (or snow depth) estimates covering the HMA region based on passive microwave (e.g. Talaka et al., 2011; Smith and Bookhagen, 2016; Dai et al., 2017; Pulliainen et al., 2020) or active microwave measurements (Lievens et al., 2019), with machine learning approaches employed to improve the accuracy in SWE estimation (e.g. Ahmad et al., 2019). Alternatively, satellite observed snow covered area products can also provide valuable information in SWE estimation. For example, fractional snow-covered area (fSCA) products are used in SWE reconstruction methods to improve the estimates of SWE over Indus and Amu Darya, by calculating snowmelt backward from melt-out to peak SWE timing using satellite-observed snow disappearance rates (Bair et al. 2018; 2020). In addition, data assimilation (DA) approaches that explicitly merged snow observations with modeling are effective in providing more realistic SWE estimates and reducing SWE uncertainties especially over the mountains (Xue et al. 2019; Largeron et al., 2020): both JRA-55 and ERA5 products assimilate ground snow depth and satellite retrieved snow cover observations; GlobSnow (Talaka et al., 2011; Pulliainen et al., 2020) products assimilate passive microwave retrieved SWE along with ground snow depth observations to provide SWE and snow extent estimates, while mountain areas with high terrain complexity are masked out. These are promising approaches to improve the accuracy in SWE estimates over HMA, yet currently there is still a need for large scale SWE datasets at higher resolution, over a longer period and covering mountainous areas in this region.

To better understand the spatiotemporal pattern and variability in seasonal snowpack over HMA, the so-called High-Mountain Asia Snow Reanalysis (HMASR; Liu et al., 2021) dataset is used herein to characterize the seasonal snow climatology and variability over HMA. The dataset covers the joint Landsat-MODIS era between Water Years (WYs) 2000 to 2017 (which will be extended to present in later versions) and was developed as part of the NASA High-Mountain Asia Team (HiMAT) activities. HiMAT is a multi-investigator effort in developing new datasets to understand cryosphere variability over HMA (Osmanoglu et al., 2017). The HMASR dataset provides daily estimates of SWE, fSCA and other snow variables, at a 16 arc-second (~500 m) resolution. SWE estimates are derived by assimilating fSCA from Landsat and

MODIS platforms using a previously developed snow reanalysis framework (Margulis et al., 2019), where the method has been shown in previous applications to provide realistic SWE estimates over mountainous domains in the Sierra Nevada (Margulis et al., 2016) and Andes (Cortés and Margulis, 2017). The HMASR aims to fill the spatiotemporal gaps in existing SWE datasets and allow for better characterization of the distribution and changes in seasonal snow storage, and provide insights into the hydrologic cycle and water availability over HMA. Using this dataset, the spatial distribution of SWE climatology is examined at annual and seasonal scales over the HMA region, covering the highest mountain ranges and the Tibetan Plateau in Asia. Integrated SWE volumes over the full HMA domain and over the major river basins (e.g., Syr Darya, Amu Darya, Indus, Ganges-Brahmaputra, Yangtze, Yellow), and their variation with elevation, are also quantified in this work. The following scientific questions are addressed herein:

1) How is seasonal snow distributed spatially across the major watersheds of HMA?

2) What is the seasonal and interannual variability in amount of snow storage over HMA?

3) How is the amount of snow distributed across elevation, and how does it vary under different climate conditions?

## 2 Data and method

This section describes the data and methods used in this study. Section 2.1 introduces the study domain, including the major river basins and mountain ranges in the region. Section 2.2 and 2.3 provide a brief description of the reanalysis method, input data and models used in the development of the HMASR. Finally, a non-seasonal snow and ice mask applied to mask out semi-permanent snow and ice for the assessment of seasonal snow is explained in Sect. 2.4.

### 2.1 HMA domain

The HMA domain used in this work is bounded from 27° N to 45° N, and from 60° E to 105° E (Fig. 1), covering the highest mountain ranges and plateaus (Tien Shan, Pamir, Hindu Kush, Karakoram, Himalayas, and Tibetan Plateau), as well as the headwaters of the main river basins (Syr Darya, Amu Darya, Indus, Ganges-Brahmaputra, Yangtze, and Yellow). Winter westerlies and the summer monsoon are the major moisture sources in this region, significantly influencing the spatiotemporal patterns in snowfall and glacier mass balance. More specifically, the northern and western HMA is dominated by westerlies and receives abundant winter snowfall, while the southern and eastern HMA is dominated by the Indian monsoon from June to September and receives a considerable amount of summer snowfall; the eastern edges of HMA are affected by the East Asia monsoon but with limited impact (Bookhagen and Burbank 2010; Yao et al., 2012; Bolch et al., 2019). Note that in HMASR, outputs are provided for each regular 1° by 1° latitude-longitude tile (within which a regular computational grid of 16 arc-second is used), and tiles with a tile-averaged elevation above 1500 m were selected and processed in the dataset (Fig. 1). This tile-average threshold (1500 m) was chosen conservatively to capture the vast majority of seasonal mountain snow over HMA, avoid running a large number of tiles with negligible snow, and reduce the

125 computational load in product development. We acknowledge this threshold might exclude snow in some areas of the domain (e.g. northern HMA) and anticipate this threshold to be relaxed or removed in future versions of this product.

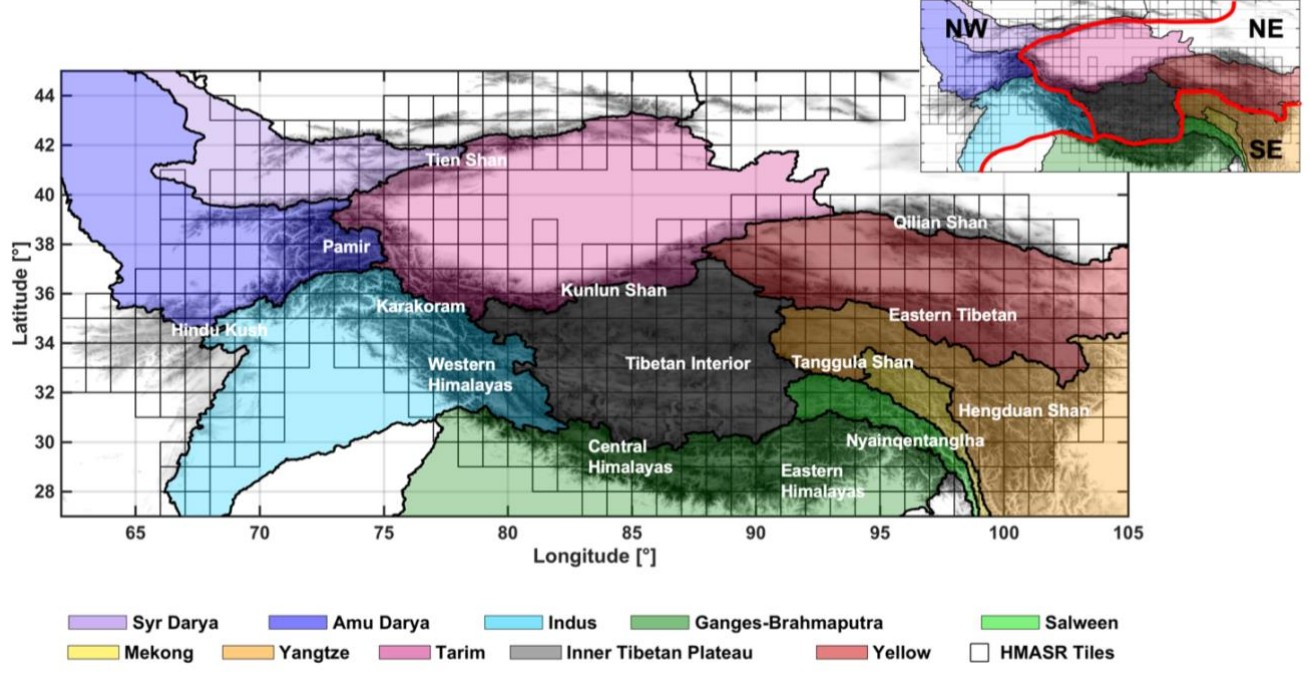

**Figure 1: Map of HMA domain with HMASR tiles marked with black boxes. Major watersheds are delineated and colored on the map based on HydroSHEDS (Lehner et al., 2008). Major mountain ranges are labeled with reference to Bolch et al. (2019). A**
130 **division of the HMA domain into Northwestern (NW), Northeastern (NE) and Southeastern (SE) sub-regions, which are used for descriptive purposes in this study, is shown in the inset.**

For convenience in presenting results herein, the HMA domain was divided into three large subregions, namely the Northwestern (NW), Southeastern (SE) and Northeastern (NE) subregions (Fig. 1). Major river basins are identified in each subregion, namely those located in NW (Syr Darya, Amu Darya and Indus), in SE (Ganges-Brahmaputra, Salween, Mekong
and Yangtze), and in NE (Tarim, Inner Tibetan Plateau and Yellow). Similarly, the major mountain ranges are also identified in each subregion, namely those located in the NW (e.g. Tien Shan, Pamir, Hindu Kush, Karakoram, western Himalayas), SE (e.g. central and eastern Himalayas, Nyainqentanglha, Tanggula Shan and Hengduan Shan), and NE (e.g. Kunlun Shan, Tibetan Interior, Eastern Tibetan and Qilian Shan), and are labelled in Fig. 1.

**2.2 Snow reanalysis scheme**

A previously developed snow reanalysis methodology (Margulis et al., 2019) is employed in deriving the HMASR. For brevity, only the key details are repeated here. Prior model estimates are obtained via the coupled Simplified Simple Biosphere model, version 3 (SSiB3; Sun and Xue, 2001; Xue et al., 2003) and the Liston (2004) snow depletion curve (SDC). The SSiB3 model is used as the land surface model (LSM) in this work, which has three snow layers with vegetation

canopy and soil representations. It requires hourly inputs of forcing data (e.g. precipitation, air temperature, radiation, wind speed, humidity and pressure) and static inputs (e.g. topography, land cover, vegetation and soil type), with more details clarified in Sect. 2.3.1. The SSiB3 model provides the basic mass and energy fluxes for the snowpack based on meteorological inputs and physiographic characteristics. These fluxes are used with the Liston SDC to derive estimates of grid-averaged SWE and fSCA. Specifically, the Liston SDC assumes that the subgrid distribution of SWE follows a lognormal distribution, and is a function of grid-averaged SWE, melt, and a parameter of subgrid coefficient of variation. The SDC yields the prediction of fSCA that is compared with satellite observed fSCA and serves as the constraint in the data assimilation.

As done in many data assimilation methods, an ensemble approach is used in the snow reanalysis scheme, whereby the model generates prior estimates of snow states (i.e. SWE, snow depth, fSCA, etc.) with postulated uncertainties. Meteorological forcing inputs are bias-corrected, downscaled to the modeling grid (16 arc-second) and perturbed with uncertainty in the ensemble approach, using the methods described in Durand et al. (2008) and Girotto et al. (2014). To constrain the prior snow estimates on the remotely sensed fSCA observations, a Bayesian update is performed using the Particle Batch Smoother (PBS; Margulis et al., 2015; Margulis et al., 2019) approach. Posterior snow estimates are obtained in this update step, by more heavily weighting ensemble members that are more consistent with the batch of observed fSCA in a given water year using a Bayesian likelihood function that accounts for model-measurement misfit and measurement error. It is worthwhile to note that, the posterior ensemble mean, median and spread (or other statistics) can be obtained via the Bayesian update step. Herein, the posterior ensemble median values of SWE are described when assessing the SWE over HMA. Details of the PBS methods are described in Margulis et al. (2015; 2019), and more details on fSCA observations are provided in Sect. 2.3.2.

The lack of in situ SWE data over HMA prevents a thorough verification of the HMASR. However, previous applications of the snow reanalysis method in similarly complex terrain in the Sierra Nevada of the Western U.S. and the South American central Andes thoroughly compared reanalysis estimates vs. in situ and airborne-derived SWE data. Performance in both domains were positive relative to in situ data with values of mean error, root-mean-squared error and correlation coefficients of: ~ 3 cm, 13 cm and 0.95 for the Sierra Nevada (Margulis et al., 2016) and ~ 1 cm, 29 cm and 0.73 for the Andes (Cortés and Margulis, 2017), respectively. In Margulis et al. (2019), comparison with the Airborne Snow Observatory (ASO) SWE data in Tuolumne in the Sierra Nevada yielded similar results (mean error, root-mean-squared error, and correlation coefficients of ~ 5 cm, 23 cm, and 0.84). Here we provide the caveat that performance of the method may be degraded in parts of the HMA region due to the frequent cloud obscuring issues (see more details in Sect. 2.3.2), compared to previous work in the Sierra Nevada or Andes.

## 2.3 Input data acquisition and processing

### 2.3.1 Meteorological, topographic and land cover data

In HMASR, the prior surface meteorological inputs were obtained from MERRA-2 at its raw resolution (0.5° by 0.625° latitude–longitude), including precipitation, air temperature, solar radiation, specific humidity, surface pressure and wind speed. The uncertainty models and their parameters used to perform bias-correction and uncertainty perturbation are specified in Margulis et al. (2019) for the HMA region, except that prior ensemble precipitation is perturbed by a lognormal distribution with mean of 1.54 and coefficient of variation (CV) of 0.83 based on the results from Liu and Margulis (2019). Digital elevation model (DEM) data were obtained from the Shuttle Radar Topography Mission (SRTM, http://www2.jpl.nasa.gov/srtm/) 1 arc-second product, and aggregated to 16 arc-second (~500 m) resolution. Gaps in DEM data were filled by the Advanced Spaceborne Thermal Emission and Reflection Radiometer (ASTER) Global Digital Elevation Model (GDEM, version 2) 1 arc-second product (https://asterweb.jpl.nasa.gov/). Land cover data were obtained from the AVHRR global land cover classification dataset (Hansen et al., 2000). Forest cover information was obtained from the Tree Canopy Cover (TCC) product containing the Landsat Vegetation Continuous Fields (https://lcluc.umd.edu/metadata/global-30m-landsat-tree-canopy-version-4; Sexton et al., 2013).

### 2.3.2 fSCA data

The fSCA observations used to condition prior snow estimates were retrieved from Landsat and MODIS platforms, for their joint period of WYs 2000 to 2017 (e.g. where WY2000 corresponds to October 1, 1999 - September 30, 2000). The (nadir-viewing) Landsat-based fSCA data were obtained from Landsat 5, 7 and 8 satellites, retrieved using a spectral unmixing algorithm (Painter et al., 2003; Cortés et al., 2014), available at 30 m and every 16 days (excluding cloudy days). The (nadir-and off-nadir-viewing) MODIS-based fSCA data were obtained from the MODIS Snow Covered Area and Grain size (MODSCAG) product (Painter et al., 2009), available daily at 500 m, with a viewing angle between 0° and 55°. Jointly assimilating fSCA from both platforms provides more measurements to compensate for cloud contamination in HMA. Cloud screening and viewing angle screening were performed as illustrated in Margulis et al. (2019), and here we only clarify the key steps for brevity. Specifically, for Landsat, any image with a diagnosed cloud cover fraction of greater than 40% is excluded entirely. For MODSCAG, only 'near-nadir' pixels within an image are included and, of those, any image with a diagnosed cloud cover fraction of greater than 10% is excluded entirely. This subset of Landsat and MODSCAG images for inclusion therefore uses a conservative screening meant to exclude significantly cloud-contaminated tiles. This does not prevent errors of omission/commission in cloud/snow identification, but is meant to mitigate cloud impacts by not including those deemed significantly cloudy. It should also be noted that the snow reanalysis method used herein is less susceptible to errors of omission/commission when compared to SWE reconstruction methods (e.g. Bair et al., 2020) that interpolate between fSCA measurements to estimate ablation rates. Instead, the snow reanalysis fitting of fSCA measurements is more akin to a least-squares type fit where measurement errors are accounted for in the framework. This

mitigates the propagation of errors of omission/commission compared to SWE reconstruction techniques. For images that passed through the cloud/viewing angle screening, cloudy pixels within each image were further excluded through internal cloud masks. After screening, both Landsat and MODSCAG images were aggregated to the same modeling resolution (16 arc-second).

No large systematic differences were seen when examining fSCA across different Landsat sensors, while substantial differences were found in same-day fSCA images between Landsat and MODSCAG (after screening and aggregation). To reconcile the inconsistency between products, a cumulative distribution function (CDF)-matching method was applied pixel-wise to statistically match MODSCAG images with Landsat images. Based on the analysis in Margulis et al. (2019), we specify a measurement error standard deviation (10% of Landsat fSCA; 15% of CDF-matched MODSCAG fSCA) in the

reanalysis to represent retrieval error/uncertainty.

It is worthwhile to note that the fSCA data availability is significantly affected by cloud contamination in some areas of HMA region, especially during the monsoon season (June-September) where fSCA measurements are limited over regions such as the Himalayas (Fig. 2). The lack of abundant fSCA data can be a potential limitation in assimilating fSCA observations for these monsoon-affected regions, and therefore leads to higher uncertainty and less constrained posterior

SWE estimates (i.e. where in the limit of no available observations, the posterior will, by construct, equal the prior estimate).

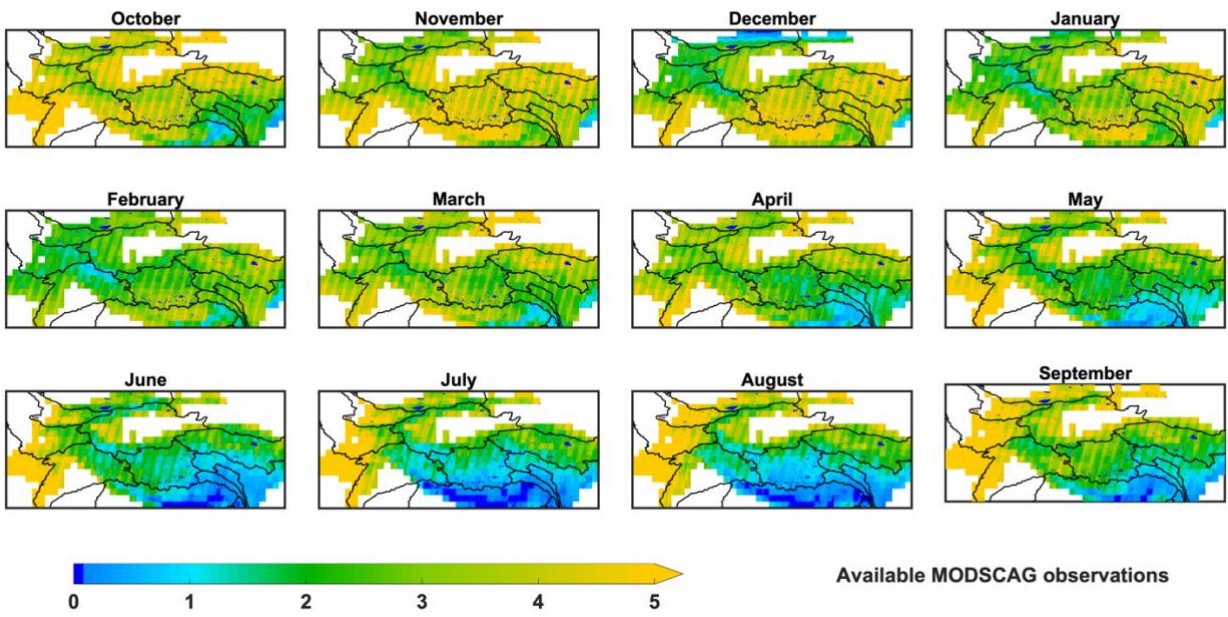

**Figure 2: Monthly total number of available (near-nadir) MODSCAG measurements averaged over 18 years, with cloud and viewing angle screening. Landsat measurements supplement these MODIS-derived measurements. The dark blue color is used to distinguish pixels with zero MODSCAG measurements.**

## 2.4 Non-seasonal snow and ice mask

A significant fraction of HMA is covered by glacier or semi-permanent snow owing to its extremely high elevation. Thus, it is important to distinguish seasonal vs. non-seasonal snow over land or glacier surfaces. In particular, the reanalysis method used in the development of the HMASR is best suited for seasonal snow characterization, because it relies on the signal between fSCA depletion time series and SWE via the LSM-SDC model. Hence, those pixels where there is not a full melt-out of snow are expected to be potentially erroneous. So, while estimates are generated at all pixels in the domain, the aim to focus on seasonal snow requires masking out semi-permanent snow and ice. Glacier inventories from the Global Land Ice Measurements from Space (GLIMS; Raup et al., 2007) and the Randolph Glacier Inventory (RGI; Pfeffer et al., 2014; RGI Consortium, 2017) have been employed in previous studies to exclude glaciers from snow modeling domains (e.g. Wrzesien et al, 2019, Smith and Bookhagen et al., 2018). Other studies such as Mudryk et al. (2015) and Mortimer et al. (2020) excluded glaciers based on estimates from the MERRA land fraction mask. Armstrong et al. (2019) applied the MODIS Persistent Ice (MODICE; Painter et al., 2012) algorithm to derive a minimum snow and ice mask based on the MODSCAG product, and used it to distinguish seasonal snow from glaciers or persistent snow.

Herein a combination method was used to exclude the non-seasonal snow and ice pixels in HMASR, based on: 1) a glacier mask derived from GLIMS to identify glacierized pixels and 2) a persistent snow mask derived from the HMASR dataset itself. We acknowledge that RGI dataset may be more appropriate to use than GLIMS, as it obtains glacier outlines around 2000 while GLIMS obtains those from a larger date range. To be more specific on the second mask, pixels with a significant amount of persistent snow were identified, by comparing the annual minimum SWE at a particular pixel to its annual maximum SWE in each year. If the minimum SWE exceeds 10% of the maximum SWE for more than once out of the 18 years, the pixel is considered to be a persistent snow pixel to be masked out in the computation of seasonal snow estimates. The derived glacier and persistent snow masks are combined into a non-seasonal snow and ice mask, which is applied when presenting the spatiotemporal patterns of seasonal SWE in the following section.

## 3 Results and discussion

The HMASR dataset is designed to provide a reliable and consistent SWE product that can be used for assessing the spatiotemporal distribution of seasonal SWE over the recent remote sensing record. To present an overall assessment of seasonal snowpack variability in the HMA region using the HMASR dataset, the results are organized as follows: 1) the spatial distribution of seasonal snowpack climatology, at annual peak and seasonal scales; 2) the temporal distribution of seasonal snowpack volume at basin and domain-wide scales; 3) the elevational distribution of seasonal snowpack storage at HMA-wide and basin scales.

**3.1 Spatial distribution of seasonal SWE climatology**

The spatial distribution of SWE is valuable in assessing the regional water storage. Given the strong seasonal signature of snowpack processes over much of the domain, the pixel-wise peak SWE is a useful metric to quantify the distribution of the maximum amount of snow water mass held in the seasonal snowpack within a given water year. Hence, the spatial distribution of peak SWE (Sect. 3.1.1) and the associated timing (Sect. 3.1.2) are examined in this section, with seasonal evolution of SWE averaged over fall, winter, spring and summer also assessed (Sect. 3.1.3).

**3.1.1 Peak seasonal SWE climatology**

The climatology (18-year average) of pixel-wise peak SWE over the HMA region is depicted in Fig. 3, where Fig. 3a presents only the results for seasonal snow pixels (where non-seasonal snow and ice pixels have been masked out). Fig. 3b presents the results for all pixels for illustration (where significantly higher amounts of SWE shows up in non-seasonal snow and ice mask pixels, corresponding to glaciers or permanent snow), where the non-seasonal snow mask covers ~4.7% of the 265   domain area. The non-seasonal SWE values (Fig. 3b) are expected to be unreliable because the initial conditions for SWE at those locations at the beginning of the dataset are unknown and the lack of full melt-out makes the relationship between fSCA depletion and peak SWE much less direct.

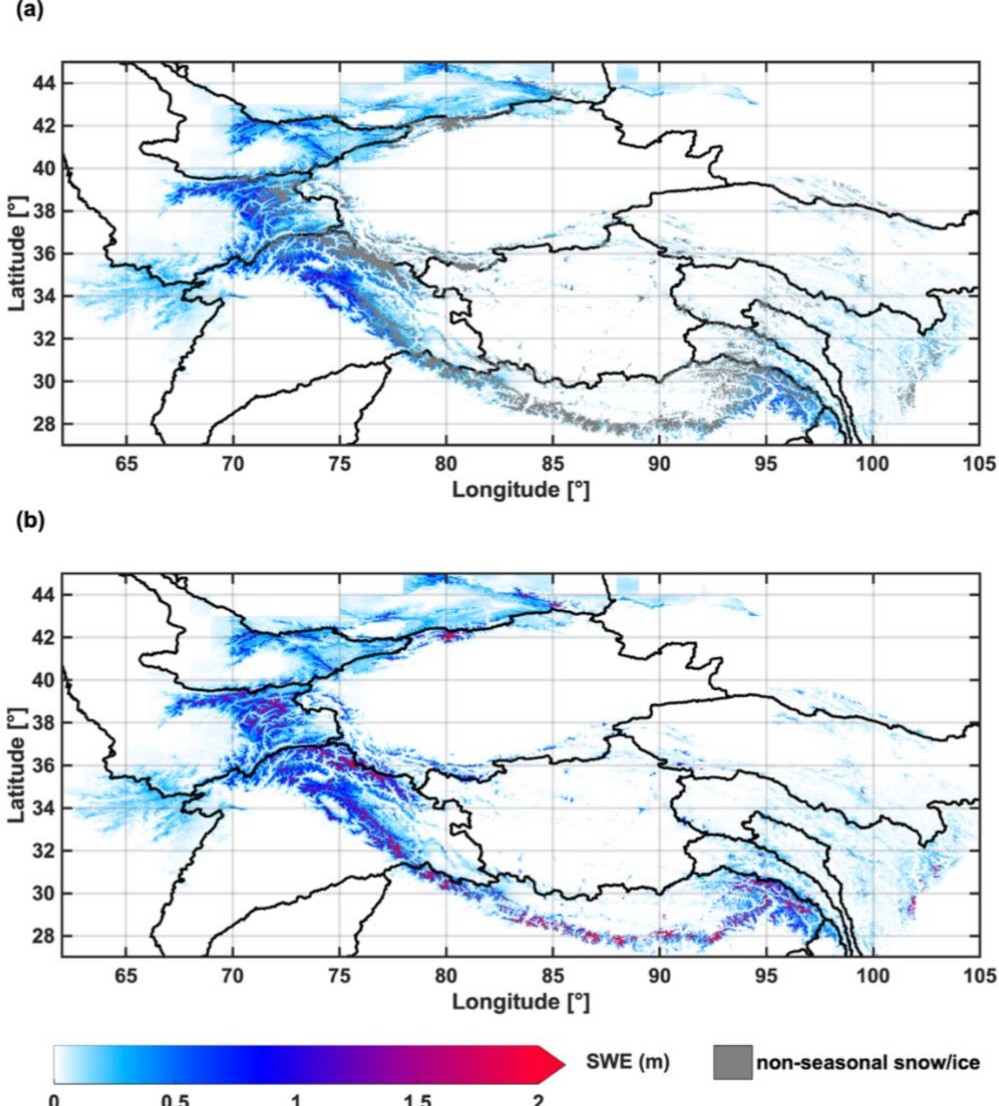

**Figure 3 (a): Map of pixel-wise peak seasonal SWE climatology, with non-seasonal snow and ice pixels masked out (grey). (b): Map of pixel-wise peak seasonal SWE climatology, without masking of non-seasonal snow and ice pixels for reference.**

In general, seasonal snow is most abundant in the NW region that is directly exposed to westerlies (Fig. 3a). Among the northwestern mountain ranges, the highest climatological peak SWE values are found in Pamir, Karakoram and the western Himalayas, with more than 1 m of peak SWE estimated. A significant amount of peak SWE is also estimated in Tien Shan and Hindu Kush, showing peak SWE values of 1 m or less in Tien Shan, and 0.5 m or less in Hindu Kush in general (Fig. 3a). The estimates of peak SWE values in Hindu Kush are consistent with measurements and SWE reconstruction estimates from Salang Pass in Afghanistan (35N, 69E, elevation 3366 m) that has records of snow (Bair et al., 2018). The non-seasonal

snow and ice is most notable in Karakoram but also evident in a few locations over the Pamir, Tien Shan and western Himalayas (Fig. 3b).

In contrast, seasonal snowpack is less abundant in the SE HMA (Fig. 3a), in part because it receives much of its precipitation in summer from the Indian and East-Asia monsoons, while the winter westerlies have minimum impact. Shallow snowpack exists over the Hengduan Shan and Tanggula Shan, with low values of SWE estimated (less than 0.2 m). For the Himalayas and Nyainqentanglha mountain ranges (Fig. 1), which exhibit extremely high elevation and receive significant summer precipitation from the monsoons, high values of SWE are estimated in some locations (Fig. 3b). However, those locations are largely masked out herein through the non-seasonal snow and ice mask (Fig. 3a), because the fSCA observations are persistently high throughout the year (no observed melt-out), show irregular temporal patterns without a clear accumulation-depletion cycle (non-seasonal), or are obscured by clouds between June-Sept. (insufficient measurements), any of which can contribute to estimates of SWE that are less constrained due to cloud screening with potential errors of omission or commission in fSCA estimation.

The least abundant seasonal snowpack is estimated in the NE (Fig. 3a), where SWE is only notable over a few mountain ranges such as the Qilian Shan, Kunlun Shan and Eastern Tibetan mountains. Despite their high elevations, most of the NE areas are snow-free or only have shallow and intermittent snow as a result of being further away from the primary atmospheric moisture sources.

Previous studies have also examined the spatiotemporal distribution in seasonal snowpack, regarding SCA (e.g. Pu et al., 2007; Basang et al., 2017), snow depth and SWE (e.g. Terzago et al., 2014; Bian et al., 2019; Orsolini et al., 2019), and the overall finding is that most existing datasets present consistent spatial patterns at large scales (e.g. regional) but differ greatly in the magnitudes of SWE and snow depth, which implies large uncertainties in snow mass estimates over this data scarce region. Similarly, HMASR exhibits coherent spatial patterns compared to these previous efforts, yet the magnitudes of SWE still show significant variability. A more comprehensive analysis of HMA SWE between multiple products will be addressed in an upcoming intercomparison paper using HMASR.

### 3.1.2 Peak seasonal SWE timing

The timing of peak seasonal SWE occurrence is associated with climatological (e.g. precipitation) and topographic (e.g. elevation) factors, and therefore shows significant heterogeneity over HMA. Figure 4 depicts the pixel-wise peak SWE day of water year (DOWY) climatology map. Highly intermittent snow pixels were excluded, as well as permanent snow and ice pixels via the non-seasonal snow and ice mask. Peak SWE generally occurs between DOWY 100 and DOWY 250 for seasonal snow. Specifically, the date of peak SWE timing is characterized spatially by a median of DOWY 169 (March 18th) and an interdecile range between DOWY133 (February 10th) and DOWY 217 (May 5th), as shown in Fig. 4. However, the peak SWE DOWY shows a bimodal distribution (Fig. 4, inset) with the earlier peak centered on DOWY 145 and the later peak centered on DOWY 192.

For those mountain ranges in the NW, the northern and western mountain slopes of Tien Shan, the western foothills of Pamir, the entire Hindu Kush, as well as the foothills of the western Himalayas, all have relatively early peak SWE occurrences between February 10th and March 18th (Fig. 4). In contrast, the southern mountain slopes of Tien Shan, the majority of the Pamir, Karakoram, and western Himalaya, show a relatively late peak SWE occurrence between March 18th and May 5th (Fig. 4). For those mountain ranges in the SE and NE, the peak SWE occurrence dates are more diverse (Fig. 4). In the SE, the central and eastern Himalayas, Nyainqentanglha, and Hengduan mountains generally have later peak SWE occurrences (between March 18th and May 5th), except in the southern foothills, where peak SWE tends to occur earlier (between February 10th and March 18th). In the NE, the eastern Tibetan mountains show the earliest peak SWE occurrence dates (before February 10th), while the Qilian Shan and Kunlun Shan show the latest peak SWE occurrences (after May 5th).

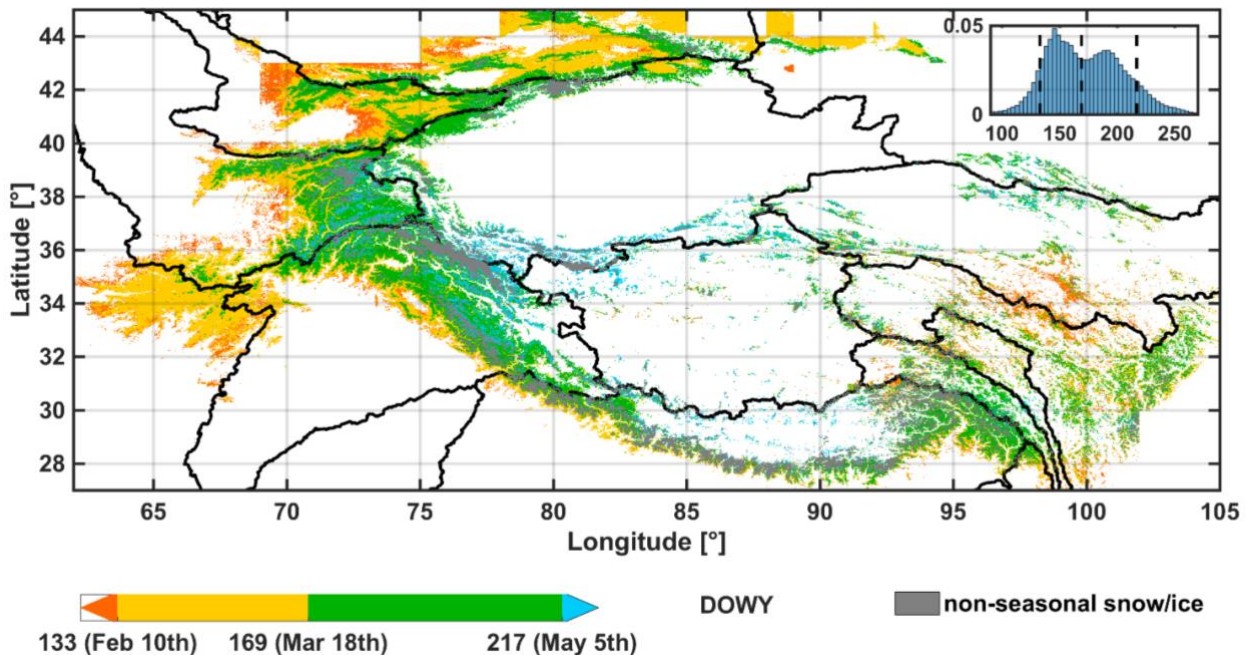

**Figure 4: Map of pixel-wise peak seasonal SWE DOWY climatology, with non-seasonal snow and ice pixels masked out (grey). The inset figure is the histogram of peak SWE DOWY. The three dates labeled in the colorbar (DOWY 133, DOWY 169 and DOWY 217) correspond to the 10th, 50th and 90th percentile in the DOWY distribution, and are marked with vertical dashed lines in the inset histogram.**

### 3.1.3 Seasonal SWE evolution

The spatial patterns of seasonal evolution of SWE, averaged over SON (September, October, November), DJF (December, January, February), MAM (March, April, May), JJA (June, July, August), are shown in Fig. 5. As expected, higher SWE amounts are generally found in winter (DJF) and spring (MAM), while lower SWE amounts are found in summer (JJA) and fall (SON). Throughout the year, mountains in NW hold the maximum amount of SWE compared to other regions. In SON, the entire HMA region exhibits minimal SWE magnitudes (0.1 m or below) and most regions are snow free (Fig. 5). During

this period, SWE starts accumulating in the Tien Shan, Pamir and western Himalayas that are directly facing the westerlies.

SWE is also evident in Nyainqentanglha and Hengduan Shan that are associated with the summer monsoons. In DJF, both the overall magnitude and extent of SWE grow significantly, with mean SWE values up to 0.5 m found in Tien Shan, Pamir and western Himalayas (Fig. 5). The magnitude of SWE grows even larger in MAM, with up to 1 m SWE values estimated in the western HMA mountains, and up to 0.5 m SWE values estimated in Nyainqentanglha and the eastern Himalayas. Meanwhile, the extent of SWE shrinks significantly during MAM in the Hindu Kush and Tien Shan due to the weakened

westerlies in spring. In JJA, both the magnitude and extent of SWE drop dramatically over most of the domain, with some exceptions of more persistent snowpack (with up to 0.3 m SWE) still evident in the Pamir, Karakoram and Nyainqentanglha, where snow melts out slower than the surrounding regions.

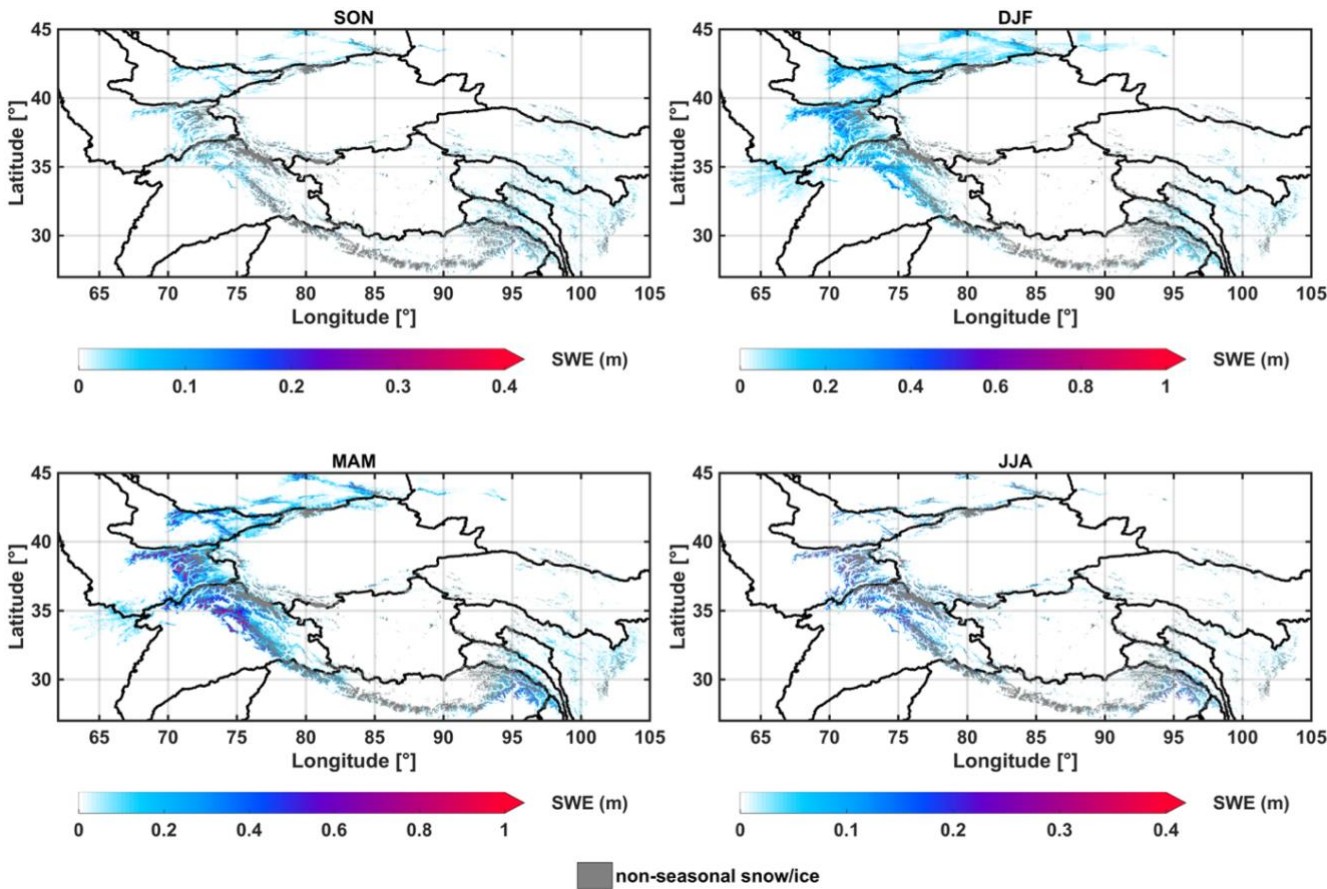

**Figure 5: Map of mean seasonal SWE climatology in SON (September, October, November), DJF (December, January, February),**
**MAM (March, April, May) and JJA (June, July, August), with non-seasonal snow and ice pixels masked out (grey).**

## 3.2 Temporal distribution of seasonal SWE

Despite the significant literature on seasonal snowpack in this region, quantification of the regional scale SWE volume is more difficult to obtain, partly due to the large uncertainties in SWE estimation over this region. In this section, the temporal variations in integrated seasonal SWE volumes across the major river basins are quantified, with the climatology presented
in Sect. 3.2.1, and the interannual variations illustrated in Sect 3.2.2.

### 3.2.1 Climatology of seasonal SWE

The climatology of the seasonal cycle in SWE volumes that are integrated across HMA and its major river basins are quantified and presented in Fig. 6, with the key statistics of annual peak SWE volumes (peak of the annual time-series) summarized for the entire HMA region and each basin in Table 1. Note again that the non-seasonal snow and ice mask has
been applied when calculating the aggregated SWE volumes.

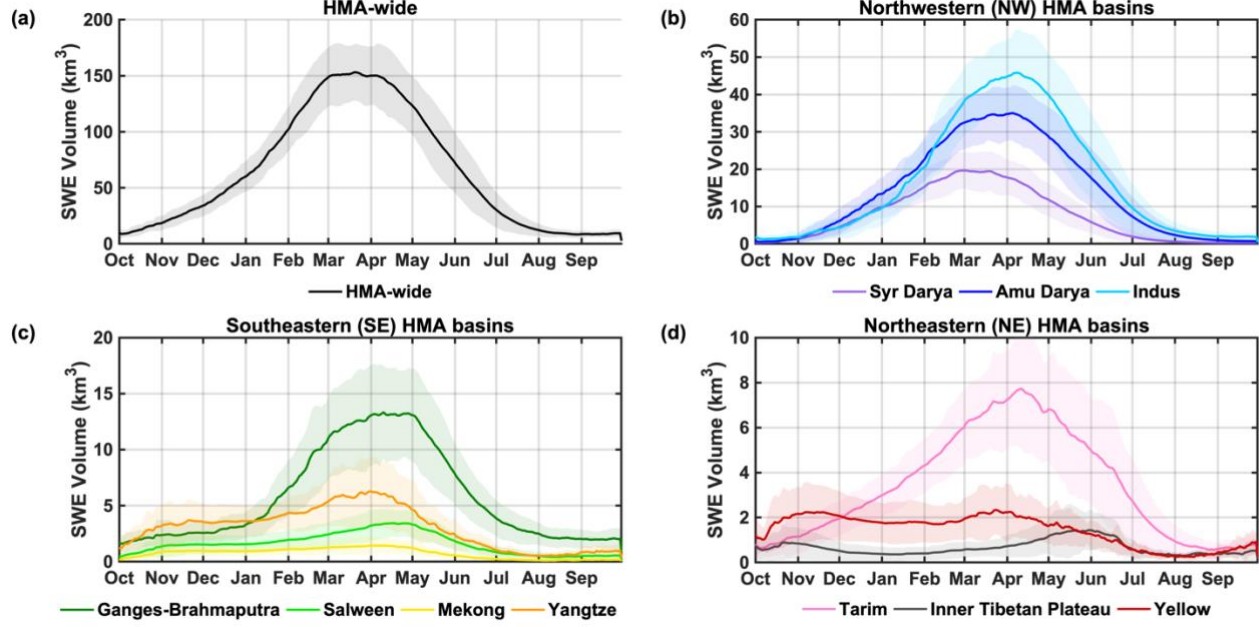

**Figure 6: Climatological (18-year average; solid line) daily time-series of seasonal SWE volumes, aggregated to a) HMA-wide, and basins in the (b) Northwestern (NW), (c) Southeastern (SE), and (d) Northeastern (NE) subregions. The shaded area represents +/- 1 standard deviation around the climatological mean (i.e. representing a metric of interannual variation about the mean).**


**Table 1: Summary statistics for HMA-wide and basin-scale annual peak SWE volume (peak of annual time-series) assessed from the 18-year HMASR.**

| Region | Basin name | Climatology | | | Interannual Variability | | | | |
| --- | --- | --- | --- | --- | --- | --- | --- | --- | --- |
| | | Mean (km³) | Total of sub-regional mean (km³) | Standard deviation (km³) | Coefficient of variation | Max (km³) | Min (km³) | Max year | Min year |
| HMA-wide | HMA-wide | **162.57** | **162.57 (100%)** | **26.53** | **0.16** | **227.12** | **114.10** | **2005** | **2001** |
| Northwestern (NW) basins | Syr Darya | 21.16 | **107.42 (66%)** | 5.19 | 0.25 | 29.88 | 13.61 | 2010 | 2000 |
| | Amu Darya | 37.31 | | 7.10 | 0.19 | 48.31 | 25.92 | 2017 | 2008 |
| | Indus | 48.95 | | 10.27 | 0.21 | 63.97 | 23.71 | 2009 | 2001 |
| Southeastern (SE) basins | Ganges-Brahmaputra | 15.59 | **29.49 (18%)** | 3.84 | 0.25 | 25.40 | 10.69 | 2005 | 2009 |
| | Salween | 3.97 | | 1.23 | 0.31 | 6.71 | 2.21 | 2005 | 2002 |
| | Mekong | 1.92 | | 0.77 | 0.40 | 3.56 | 1.02 | 2000 | 2004 |
| | Yangtze | 8.02 | | 2.97 | 0.37 | 14.79 | 3.32 | 2005 | 2015 |
| Northeastern (NE) basins | Tarim | 8.78 | **14.78 (9%)** | 2.45 | 0.28 | 12.92 | 4.81 | 2017 | 2007 |
| | Inner Tibetan Plateau | 2.35 | | 0.99 | 0.42 | 4.98 | 0.57 | 2013 | 2004 |
| | Yellow | 3.65 | | 1.25 | 0.34 | 6.10 | 1.76 | 2005 | 2004 |

The HMA-wide SWE volume is presented in Fig. 6a, and the 18-year average of annual peak SWE volume is found to be 162.57 km³ (note this is higher than the peak value in Fig. 6a, as it is the direct average of 18-year maxima rather than averaged across DOWY). The climatological peak SWE volume was further assessed in each subregion (i.e. within NW, NE and SE), and compared against that over the entire HMA (Table 1). The results show the highest peak SWE volume occurs in NW basins (107.42 km³, ~ 66% of domain-wide total), followed by SE basins (29.49 km³, ~18%), and NE basins (14.78 km³, ~ 9%), which is coherent with the spatial pattern shown in Fig. 3a. Note that around ~7% of HMA-wide SWE volume falls in the regions outside of the watersheds examined (mainly in the northmost regions shown in Fig. 1), which is why these basin-scale quantities do not sum up to 100% of the HMA-wide totals.

For the NW basins, the maximum amount of SWE volume is found in the Indus basin, followed by Amu Darya and Syr Darya (Fig. 6b). The seasonality of basin-scale SWE in NW displays similar features to the HMA-wide SWE, with snow accumulating from October to March/April and depleting until the end of the WY. Meanwhile, the peak SWE volume is found to occur earlier and disappear faster in the Syr Darya basin, followed by Amu Darya and Indus. This is potentially attributed to their geographic locations, where Syr Darya is located further north and only affected by the winter westerlies;

Indus is located further south and is partially affected by the summer monsoons.

For the SE basins, higher SWE volumes are found in Ganges-Brahmaputra, followed by Yangtze, Salween and Mekong (Fig. 6c). It is worthwhile to note that Ganges-Brahmaputra has an average peak SWE volume of 15.59 $km^3$, with an average carry-over SWE volume of around 2 $km^3$ at the end of the WY (Fig. 6c). This amount of carry-over SWE volume in Ganges-Brahmaputra is a result of the facts that 1) its mountain ranges (Himalayas) have higher elevation than those in other basins,

and 2) based on the non-seasonal snow and ice criterion (Sect. 2.4), a carry-over SWE within 10% of the maximum SWE in each year is allowed. Meanwhile, the seasonality in basin-scale SWE over SE is distinct across basins, e.g. the Ganges-Brahmaputra and Salween shows more unimodal features (with obvious peaking in April-May), while Yangtze and Mekong shows more bimodal/uniform features, which are likely to be associated with the intermittent snowpack and summer monsoons.

For the NE basins over HMA, the overall magnitude of SWE volumes is smallest (Fig. 6d). These basins all have relatively large areas, but are mostly snow-free or covered by shallow snow as depicted in Fig. 3a. Distinct seasonal features are also observed in these basins, e.g. a unimodal seasonal cycle of SWE is found in Tarim with an obvious peak in mid-April, while the Inner Tibetan Plateau and Yellow show more uniform features that are potentially attributed to the intermittent snow, as they are further away from the moisture sources (limited influence by westerlies and monsoons).

**3.2.2 Interannual variations in SWE and timing**

In addition to the 18-year climatology of SWE volumes in Sect. 3.2.1, the interannual variations in HMA-wide and basin-scale peak SWE and its timing are further illustrated in Fig. 7-9 and Table 1. The aggregated seasonal SWE volume across HMA-wide or basin-scales are visualized in the 18-year time-series (Fig. 7), which illustrates a strong seasonal cycle and significant interannual variations in peak SWE. Over the record examined, the HMA-wide annual peak SWE volume (Table

1; Fig. 7) is found to be largest in WY2005 with a value of 227.12 $km^3$, smallest in WY2001 with a value of 114.10 $km^3$, and has a standard deviation of 26.53 $km^3$ (i.e. a coefficient of variation of 16%). Basin-scale annual peak SWE volumes also exhibit significant variations, with their standard deviations ranging from 0.77 $km^3$ (coefficient of variation of 40%) in Mekong to 10.27 $km^3$ (coefficient of variation of 21%) in Indus. Moreover, different maximum/minimum years of peak SWE are found in each basin, and are not always synchronous with the maximum/minimum years found in HMA-wide SWE

(Table 1; Fig. 7).

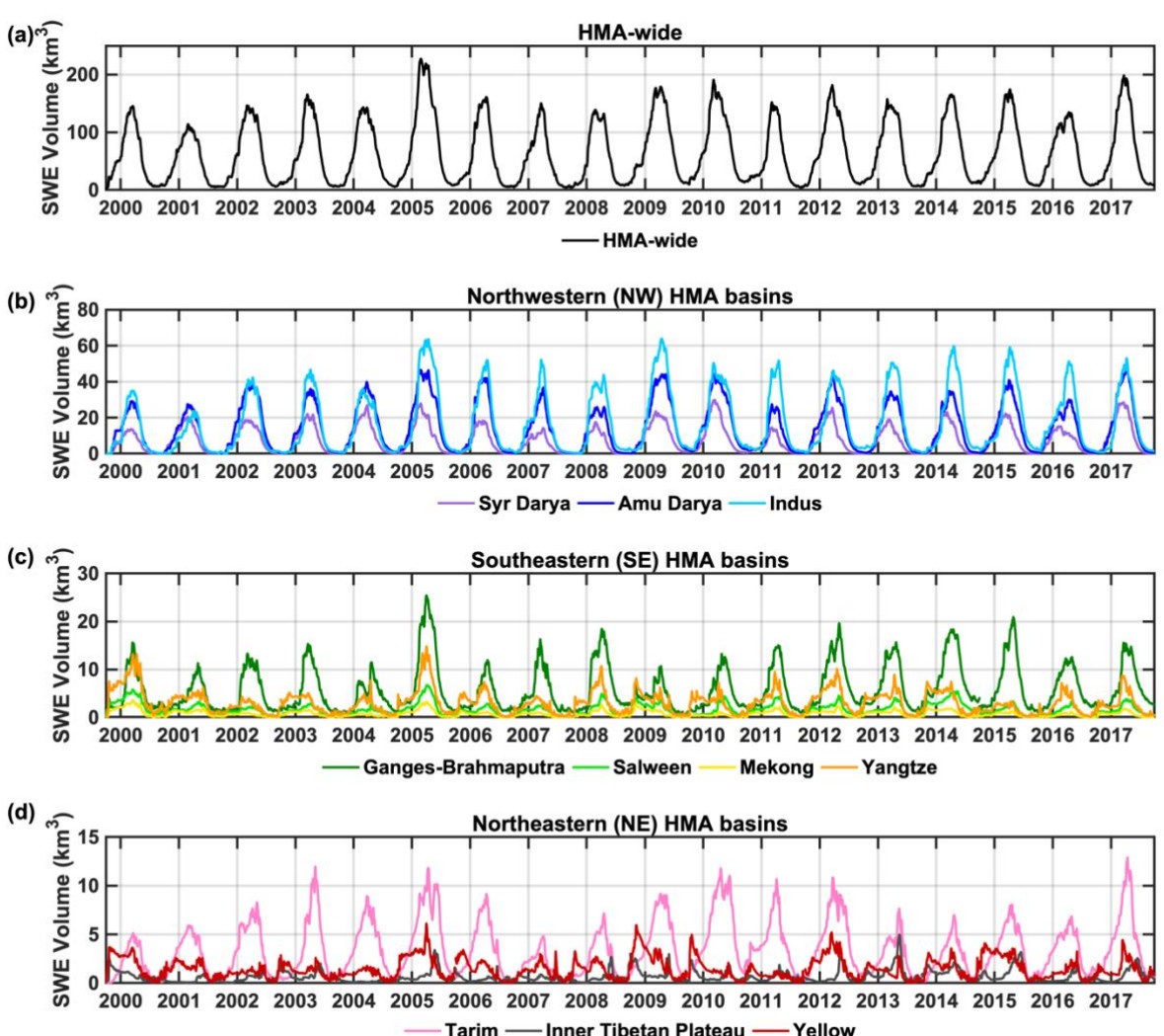

**Figure 7: Daily time-series of seasonal SWE volumes aggregated to (a) HMA-wide, (b) Northwestern (NW), (c) Southeastern (SE), and (d) Northeastern (NE) basin totals.**

When focusing on the HMA-wide seasonal cycle across different WYs (Fig. 8), it is found that snowpack quickly accumulates to over 10 km$^3$ in SWE volume during October, approaching ~50% of its peak SWE volume in January for most WYs, reaching a peak SWE volume within March and April, with an averaged timing of DOWY 168 (March 17th) when averaged over 18 WYs. After peaking, the seasonal snowpack starts depleting and declines back to ~50% of its peak volume in May and June for most WYs. Snowpack depletes to under 10 km$^3$ in SWE volume between July and September, except in WYs 2009, 2010 and 2014 that have persistent snow across the entire year. The interannual variations across WYs are evident in: 1) the variation in peak SWE volume and the peak dates, which range from 114 km$^3$ to 227 km$^3$ in volume and

from late February (DOWY 146) to mid-April (DOWY 195); 2) the variation in the temporal window where the snow storage is more than 50% of the peak SWE, that span between 3.5 months (WY2003) to 5.5 months (WY2016); 3) the variation in timing when snowpack depletes to under 10 km$^3$ in SWE volume, which ranges between July and October.

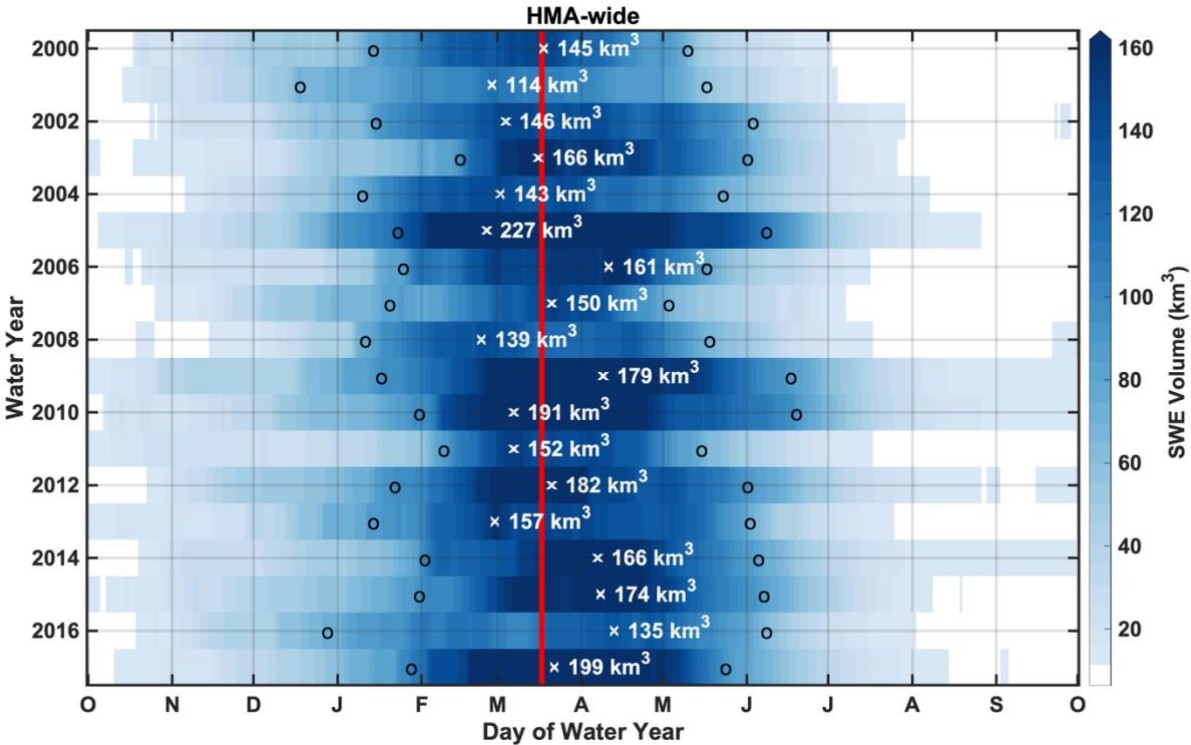

**Figure 8: Daily time-series of HMA-wide SWE volumes displayed as functions of DOWY and WY. The symbol 'x' is used to mark the date of peak SWE volume occurrence, with the corresponding peak SWE volume labeled in each WY. The symbol 'o' is used to mark the dates when 50% of the peak SWE volume is reached in each WY. The vertical red line is used to indicate the 18-year average timing of HMA-wide peak SWE volume.**

The basin-scale results (Fig. 9) show more variation compared to the HMA-wide results, with divergent peak SWE dates
across basins and across WYs. The seasonal cycle observed in NW and many other basins (Syr Darya, Amu Darya, Indus, Ganges-Brahmaputra, Tarim), are clearly influenced by winter westerlies, with SWE typically peaking around April and depleting in July – October, and that seasonality is consistent when examined across different WYs. The interannual variations in these basins are mainly reflected by 1) the overall magnitude of SWE volumes, 2) timing of snowpack occurrences/disappearances, while the peak SWE dates are closely centered around the climatological mean dates (~April).
However, different seasonal cycles are observed in the other basins (Mekong, Yangtze, Inner Tibetan Plateau and Yellow) that are more influenced by the summer monsoons, when examined across different WYs. For example, peak SWE may occur as early as October and as late as July within the same basin. The persistence of snow also varies across basins and

WYs, with SWE being either persistently high across several months, or intermittent over a short period of time. These factors explain the bimodal or uniform features in the SWE time-series and its climatology (Fig. 6-7).

It is also worthwhile to note that the average peak SWE dates are in March and April for most basins, while it is not necessarily representative for some basins (e.g. Inner Tibetan Plateau and Yellow) that have highly varying dates across years. Moreover, the average dates in many basins appear to be later than the HMA-wide average peak SWE date (March 17th), mainly because a portion of the HMA-wide SWE falls in the northmost regions that is outside of the watersheds examined (above Syr Darya), and those regions are most influenced by the winter westerlies and reach peak SWE very early

(before March 19th; Fig. 4).

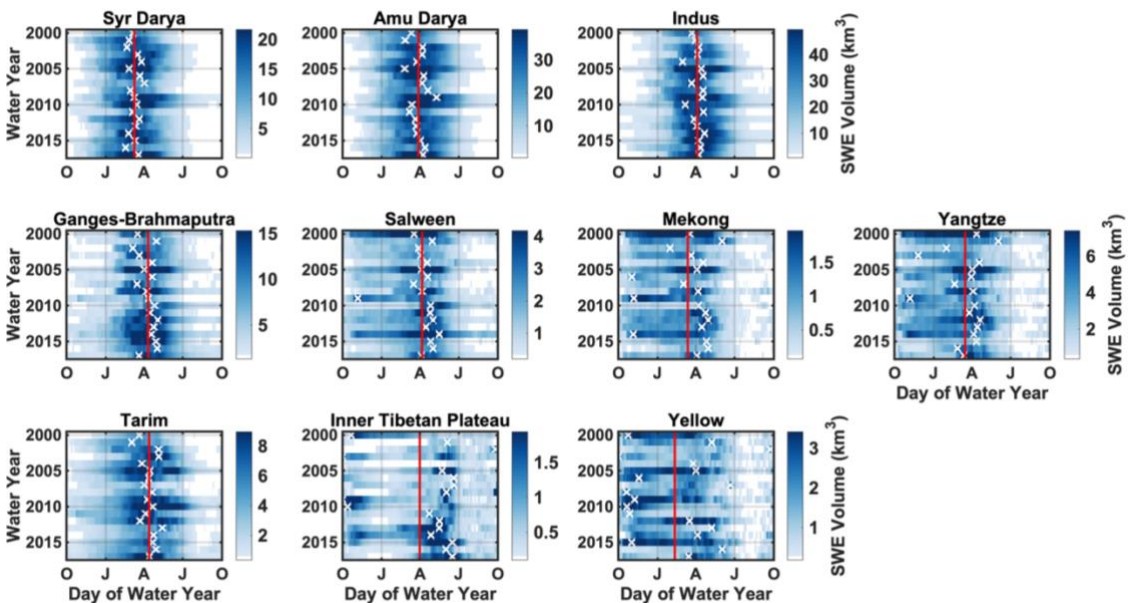

**Figure 9: Daily time-series of basin-scale SWE volumes displayed as functions of DOWY and WY. The symbol 'x' is used to mark the date of basin-scale peak SWE volume occurrence in each WY. The vertical red line is used to indicate the 18-year average timing of basin-scale peak SWE volume.**

**3.3 Elevational distribution of seasonal SWE**

The pixel-wise peak SWE distribution vs. elevation was assessed, both in terms of its 18-year averaged climatology (Sect. 3.3.1) and its variations under different climate conditions (Sect. 3.3.2). The HMA-wide domain and each basin were divided into 5-percentile elevation bins, so that the aggregated SWE volumes are calculated over comparable areas, following the method in Smith and Bookhagen (2018). The non-seasonal snow and ice pixels were removed when calculating peak

seasonal SWE volume, and its fractional areal coverage within a given elevation band is computed to assess the relative elevational contributions to total seasonal SWE volume.

### 3.3.1 Seasonal peak SWE climatology

When examining the SWE climatology over the full HMA domain (Fig. 10), the seasonal pixel-wise peak SWE volume was found to be largest at mid elevations (3000 - 4000 m), with peak SWE values occurring at elevations around 3500 m (Fig. 10a and Fig. 10b, top row). The large increase in SWE from lower to mid-elevations is indicative of orographic enhancement, where the decrease at higher elevations is indicative of moisture limitations on orographic effects and/or increasing amounts of non-seasonal snow and ice. The presence of non-seasonal snow and ice becomes evident at elevations above 3500 m, and it increases dramatically above 5000 m with a value up to 35% (Fig. 10c, top row). When assessing the cumulative fraction of SWE volume as a function of elevation, it was found that over 50% of HMA-wide seasonal SWE volume is stored at elevations above 3500 m, and less than 10% of seasonal SWE volume is stored at elevations below 2000 m (Fig. 10d, top row).

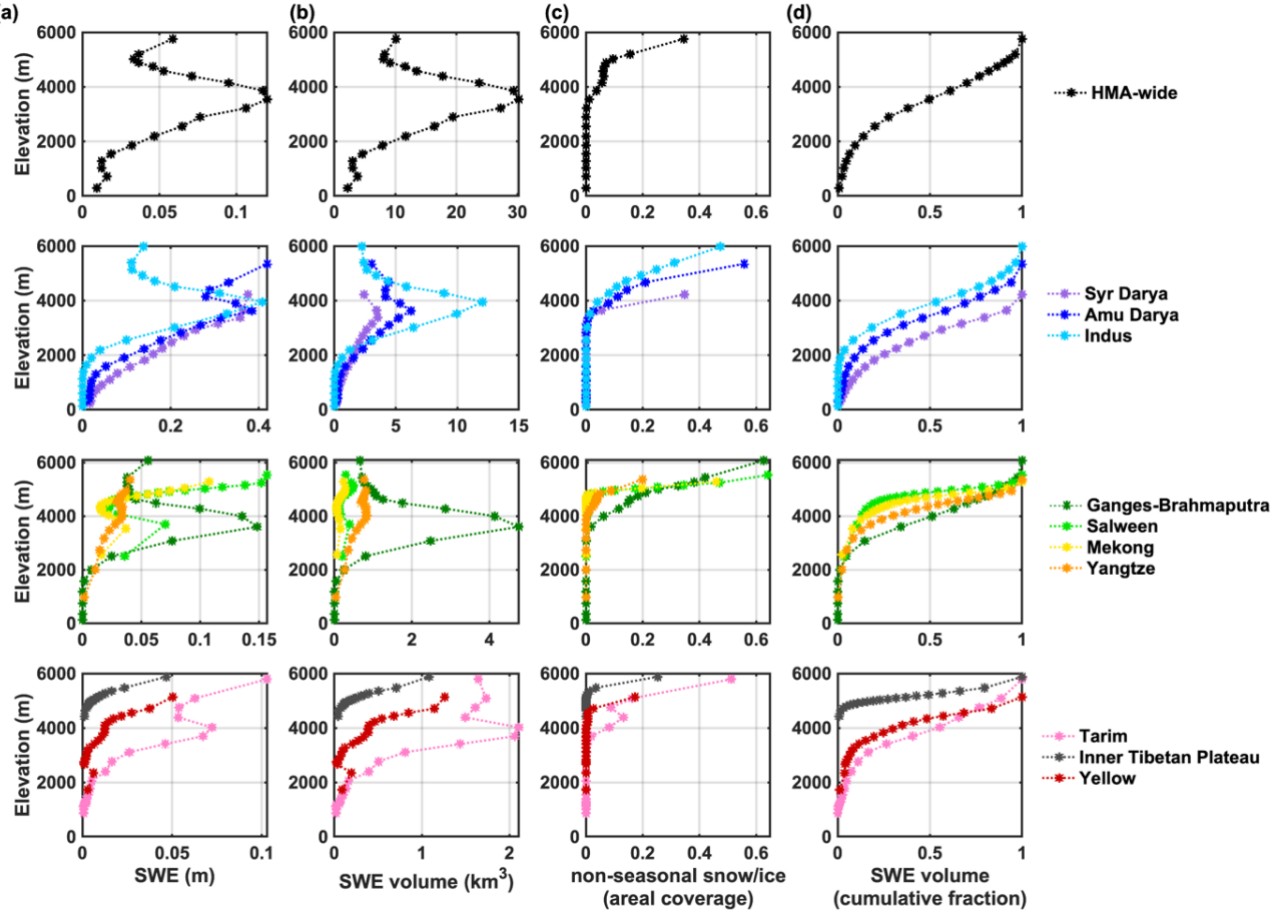

**Figure 10: Pixel-wise peak seasonal (a) SWE climatology, (b) SWE volume climatology, (c) fractional areal coverage of non-seasonal snow and ice within each elevation band, and (d) cumulative fraction of SWE volume above the specified elevation, within HMA, Northwestern (NW), Southeastern (SE) and Northeastern (NE) basins. Elevation is discretized into 5% percentile bins.**

The subregional elevational distribution of pixel-wise peak SWE climatology and its volume vary in each basin, compared to the HMA-wide results (Fig. 10). Relatively similar characteristics are observed in the NW basins (Syr Darya, Amu Darya and Indus), Tarim in NE, and Ganges-Brahmaputra and Yangtze in SE, where the pixel-wise peak SWE volumes (Fig. 10b) generally increase with elevation (below ~4000 m) and then decline with elevation (above ~4000 m), reaching their maximum values at mid elevations (3000 - 4000 m). While the SWE distribution (Fig. 10a) is generally consistent with the SWE volume distribution (Fig. 10b) within these basins, the SWE values at high elevations (e.g. ~6000 m) are large, contributing to a non-negligible amount of seasonal SWE volume at high elevations, despite the relatively high coverage of non-seasonal snow and ice (up to 60%; Fig. 10c) above 4000 m in these basins. Moreover, for the cumulative fraction of SWE volumes above a specified elevation (Fig. 10d), unique median values are found within each basin, e.g ~4000 m (Indus, Ganges-Brahmaputra, Tarim), ~3500 m (Amu Darya), or ~2800 m (Syr Darya).

Other basins in SE (Salween, Mekong) and NE (Yellow, Inner Tibetan Plateau) generally show monotonically increasing SWE and SWE volumes against elevation (Fig. 10a, 10b). These basins feature relatively small SWE volumes, and low coverage of non-seasonal snow and ice coverage (mostly under 25%; Fig. 10c) at high elevations. For the cumulative fraction of SWE volumes above a specified elevation (Fig. 10d), the median values are found at higher elevations for SE and NE (between 4000 - 5200 m) basins.

### 3.3.2 Variations under different climate conditions

The elevational distribution of peak SWE was also examined under different climate conditions (e.g. warm vs. cool years, wet vs. dry years) relative to normal conditions. Such analysis identifies whether different climate conditions affect the overall snow storage distribution across different elevations. For categorizing the different climate conditions, the HMA-wide winter precipitation and (near-surface) air temperature were used (Table 2), where winter (DJF) denotes the period from December 1st to March 1st. A "k-means" clustering analysis method (Lloyd, 1982) was used to seek classification of different climate conditions, based on the normalized winter precipitation and air temperature (subtracting the mean values and dividing by the standard deviations). The number of clusters to be classified is an input to the method; five clusters were specified in an attempt to group annual realizations into 'normal', 'wet', 'dry', 'warm' and 'cool' categories. The classified clusters are displayed in Fig. 11, where the five clusters are logically grouped and interpreted as the categories mentioned above. It should be noted that there is a slight correlation (correlation coefficient of 0.29 with a p-value of 0.25) between annual realizations of precipitation and air temperature, indicating warmer years tend to be wetter years (but statistically insignificant due to the limited number of years).

**Table 2: HMA pixel-wise peak SWE volume, winter precipitation volume and winter air temperature, with each year categorized as dry/normal/wet/warm/cool based on clustering classification.**

| WY | Peak SWE volume pixel-wise (km$^3$) | Winter | | Clustering Category |
| | | Precipitation volume (km$^3$) | Air temperature (K) | |
|------|--------|--------|--------|--------|
| 2000 | 214.08 | 251.51 | 261.65 | dry |
| 2001 | 187.97 | 204.28 | 262.21 | dry |
| 2002 | 240.66 | 309.23 | 262.60 | normal |
| 2003 | 266.22 | 330.72 | 262.71 | normal |
| 2004 | 235.36 | 313.29 | 262.31 | normal |
| 2005 | 355.55 | 422.90 | 262.14 | wet |
| 2006 | 254.80 | 318.94 | 263.04 | warm |
| 2007 | 224.51 | 303.16 | 262.43 | normal |
| 2008 | 249.79 | 316.56 | 260.73 | cool |
| 2009 | 294.83 | 339.96 | 263.19 | warm |
| 2010 | 313.81 | 331.18 | 262.53 | normal |
| 2011 | 237.69 | 294.65 | 261.54 | cool |
| 2012 | 283.24 | 282.77 | 260.77 | cool |
| 2013 | 263.14 | 337.68 | 262.01 | normal |
| 2014 | 273.41 | 292.81 | 262.19 | normal |
| 2015 | 266.62 | 330.90 | 262.71 | normal |
| 2016 | 226.62 | 258.84 | 262.48 | dry |
| 2017 | 306.40 | 351.42 | 263.68 | warm |

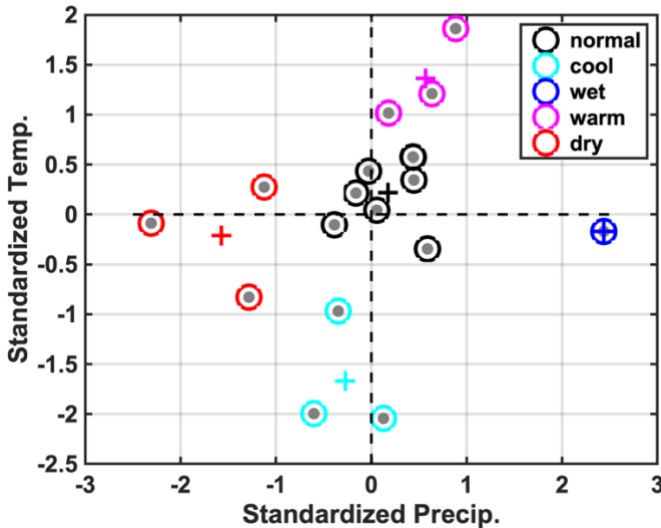

**Figure 11: Category of different climate conditions from clustering analysis, based on the normalized winter precipitation and winter air temperature. Five clusters were identified as normal, wet, dry warm and cool conditions, with the centroid of each cluster marked with the '+' symbol.**

Based on these clusters, the average pixel-wise peak SWE volumes under the same climate conditions are computed. The cluster-averaged peak SWE volume under dry, normal, and wet years are 209.6 km$^3$, 260.5 km$^3$, and 355.6 km$^3$ respectively. As shown in Fig. 12a, as expected, in spanning from dry to wet years there are marked increases in peak SWE volume over all elevations, particularly over the mid to low elevations (e.g. below 4000 m). In drier years, while Fig. 12a shows less SWE volumes across all elevations, the fractional SWE volumes are not always smallest, as shown in Fig. 12b. It can be observed that the fractional SWE volumes in dry years are smaller than those in normal years, in the low-to-mid elevations (~1500 - 3600 m). At mid-to-high elevations (~3600 - 5500 m), the dry years show greater fractional SWE volumes, compared to normal years. On the contrary, wet years show larger fractional SWE volumes below ~ 3000 m, and smaller fractional SWE volumes above ~3000 m, when compared to normal years. Such differences in the fractional SWE volumes may be due to two potential factors: 1) the dry conditions generally have less humid air, which may have accelerated evaporation and snow sublimation at lower elevations prior to peak timing; and 2) a slight shift in snowfall/precipitation towards higher elevations during drier years due to orographic effects, i.e. precipitation tends to occur at higher elevations where the moist and less humid air is cold enough to reach condensation. Note that the cluster-averaged air temperature is quite consistent under wet/dry/normal conditions, which should minimize the effect of air temperature differences on snow distribution in Fig. 12a and Fig. 12b.

Similarly, the pixel-wise peak SWE distribution under warm, normal, and cool years are examined. The cluster-averaged peak SWE volumes under warm, normal and cool years are 285.34 km$^3$, 260.47 km$^3$, and 256.91 km$^3$, respectively, with cluster-averaged air temperatures of 263.30 K, 262.44 K, and 261.01 K. It should be noted that, the warm year cluster has greater peak SWE volumes (by ~25 km$^3$) than the normal and cool years, which is reflected by the slightly higher

precipitation in the warm year cluster (Fig. 11). Therefore, we put more emphasis on the fractional peak SWE distribution

(Fig. 12d) here, to eliminate the effect of overall snow volume on snow distribution. The results indicate that warm and

normal years have very consistent distributions, when the fractional SWE volumes are examined (Fig. 12d). It is most

notable that cool years have higher fractions of snow stored at lower elevations (e.g. below 3000 m), and smaller fractions of

snow stored at mid elevations (3000 - 4000 m), compared to normal and warm years (Fig. 12d dotted lines). As the

difference in air temperature between normal and cool years is ~1.5 K, this may indicate that the low elevation snow storage

tends to shift towards higher elevations (e.g. to mid elevations) with 1.5 K of warming (from cool to normal conditions),

when the overall snow storage is the same.

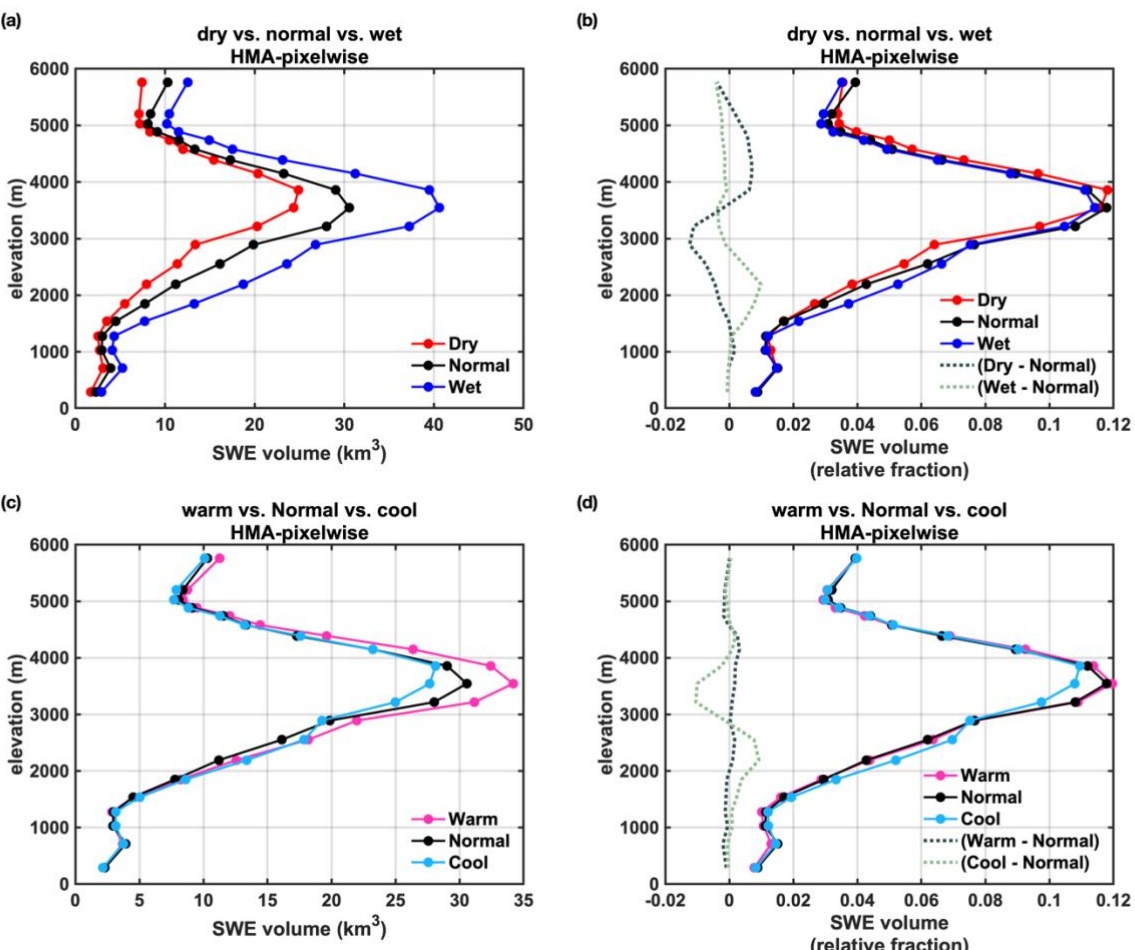

**Figure 12: Cluster-averaged pixel-wise peak SWE volume (and its relative fraction, i.e. normalized with total SWE volume) distribution vs. elevation under different climate conditions in HMA. Here (a) and (b) show the distribution under dry, normal, and wet conditions; (c) and (d) show the distribution under warm, normal, and cool conditions. Difference curves with reference to the normal condition are also provided in (b) and (d) as shown with dashed lines.**

# 4 Conclusions

A first-order spatiotemporal analysis of seasonal SWE over the HMA region is presented in this paper, using a new 18-year snow reanalysis dataset (HMASR; Liu et al., 2021). This HMASR dataset is derived based on a previously developed snow

reanalysis scheme (Margulis et al., 2019) that jointly assimilates fSCA observations from both Landsat and MODSCAG products, which has daily outputs of SWE and other snow variables, with a spatial resolution of 16 arc-second (~500 m), over the joint period of Landsat and MODIS from WYs 2000 to 2017.

This work herein used the new HMASR dataset to address scientific questions aimed at characterizing how seasonal SWE and snow storage is distributed spatially, temporally and elevationally across and within HMA. In terms of the spatial

distribution, seasonal snow is most abundant in the NW, with over 1 m of peak SWE observed over the mountain ranges. Seasonal snow is also significant in the SE, where both relatively deep snowpacks (with peak SWE values up to 1 m or above) and shallow snowpacks (with peak SWE up to 0.2 m) are found. Seasonal snow is less abundant in the NE where most areas are snow free, or only covered by shallow snowpacks (with peak SWE values below 0.2 m). The domain-wide median date of peak SWE is estimated to be March 18th with significant heterogeneity across this region, linked with

climatological drivers and topography.

When aggregating the total SWE volumes across the full HMA domain and its basins, the climatological peak seasonal SWE volume was found to be 163 km$^3$, with NW basins accounting for around 66% of that volume, followed by SE (~18%) and NE (~9%) basins. The seasonal cycle of HMA-wide SWE is depicted by snow accumulating through October to March and April, typically peaking around April and depleting in July-October. When examined at basin-scales, similar seasonality is

observed in the westerly-affected basins (e.g. in NW), while different SWE seasonality is observed in monsoon-affected basins (in SE and NE). Interannual variations in HMA-wide or basin-scale SWE are also evident, with peak SWE volumes ranging from 114 km$^3$ to 227 km$^3$ and peak dates ranging from late February (DOWY 146) to mid April (DOWY 195), when examined over the HMA-scale. The basin-scale SWE is more different from the HMA-wide SWE, where peak SWE may occur as early as October and as late as July, and are divergent across basins and across WYs.

The climatology of HMA-wide seasonal peak SWE is found to be most abundant at mid-elevations (3000 - 4000 m), with over 50% of the seasonal SWE volume stored at elevations above 3500 m. When comparing wet, normal, and dry years, we found that years with above-average amounts of precipitation causes significant overall increase in SWE volumes across all elevations. Meanwhile, a slight increase in air temperature (e.g. ~ 1.5 K) from cooler to normal years, mainly leads to an redistribution in snow storage from lower elevations to mid elevations, when the overall snow volume is the same.

This HMASR dataset is presented to augment the spatiotemporal gaps in previous SWE datasets and provide better characterization of spatiotemporal patterns in seasonal snowpack over the HMA region, especially over the mountainous areas with complex terrain where existing products tend to underestimate SWE and present large uncertainties (Wrzesien et al., 2019; Kim et al., 2021). It should prove useful in providing more insight into the role of seasonal snowpack in the

regional hydrological cycle, as a verification dataset for atmospheric and other models, and in other applications where a space-time continuous snow dataset constrained by remote sensing data is needed.

It should be noted that the reanalysis method is generally expected to work best for seasonal snow where there is a strong signature between snow disappearance and measured fSCA. Hence an important caveat is that non-seasonal snow pixels are likely to be more erroneous than the seasonal snow pixels. The use of a non-seasonal vs. seasonal snow mask is used in this paper to highlight the part of snow storage that is deemed seasonal snow. In the raw dataset, all pixels are provided and so users are free to take advantage of the non-seasonal snow estimates (with the caveat mentioned above). For the purposes of highlighting a new estimate of seasonal snow climatology in this paper, we focus on seasonal snow alone.

It is also acknowledged that the reanalysis method is best designed for non-ephemeral snow where there is a strong seasonal cycle and signal between snow disappearance and measured fSCA that can be captured at the frequency of the fSCA measurements. Hence ephemeral snow (i.e. shallow and intermittent) may not be fully captured. Finally, the accuracy of fSCA retrievals are likely not as high in the monsoon dominated parts of HMA, which in our case excludes many more Landsat/MODSCAG measurements, resulting in higher uncertainty in SWE estimation over affected sub-regions like the Himalayas. Other remote-sensing approaches (e.g., active microwave measurements) that could penetrate clouds may potentially aid in reducing the uncertainties for SWE estimation over those areas. More research can be done to address such issues and improve the accuracy of SWE estimates for those regions in the future.

*Data availability.* The HMASR dataset used in this paper, is publicly available on National Snow and Ice Data Center (NSIDC) HiMAT data repository, entitled: 'High-Mountain Asia UCLA Daily Snow Reanalysis, Version 1'. It can be accessed through https://nsidc.org/data/HMA_SR_D/ or https://doi.org/10.5067/HNAUGJQXSCVU (Liu et al., 2021). The dataset is provided as NetCDF files for each 1° x 1° tile shown in Fig. 1, available at 16 arc-second (~ 500 m) and daily resolution from WYs 2000 to 2017. Posterior estimates of other key snowpack properties (i.e. in addition to SWE) not focused on herein (e.g. snow depth, fSCA, snowmelt, sublimation, snow albedo, etc.) along with posterior forcing variables are included in this dataset. Data quality information, containing a classification mask and the non-seasonal snow/ice mask, can be found in the dataset as well. Future versions of the dataset could extend it to include other years, provide estimates at higher spatial resolutions, and better characterize uncertainties through inclusion of other meteorological forcings and other inputs to the reanalysis framework.

*Author contribution.* YL led the production of the dataset and led the analysis in the manuscript. YF contributed to the production of the dataset and contributed to the analysis in the manuscript. SM supervised the project and provided guidance. All authors contributed to writing the manuscript.

*Competing interests.* The authors declare that they have no conflict of interest.

*Acknowledgements.* We would like to thank those responsible for the creation of the datasets used in this study as well as
members of the NASA High-Mountain Asia Team (HiMAT) who helped shape the overall direction of the work. G. Cortes
is acknowledged for pre-processing some of the input data used to develop the HMASR dataset. This research was funded by
the NASA High-Mountain Asia Program Grant #NNX16AQ63G with additional support provided by NSF Grant #1641960.

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
