# Peer review of "Spatiotemporal distribution of seasonal snow water equivalent in High-Mountain Asia from an 18-year Landsat-MODIS era snow reanalysis dataset"

_The Cryosphere, 2021_

## Referee Comment (RC2)

[referee-annotated manuscript omitted]

---

## Author Comment (AC1)

We would like to thank the reviewer for the very helpful and constructive comments in suggesting improvements in our original manuscript. Below we provide point-by-point responses to the comments, where any proposed changes would be finalized during the revision of the manuscript made in the next stage.

**Reviewer # 1:**

As mentioned in my summary above, I was surprised how little additional analysis there was in this manuscript beyond exploring the dataset. Especially since the algorithm development and discussion is published elsewhere, I would like to see additional analysis here. On line 249, you say "A more comprehensive analysis of HMA SWE between multiple products will be addressed in an upcoming intercomparison paper using HMASR." Can that intercomparison not be included here? This did not seem like a very long manuscript, and understanding how the HMASR dataset compared, and likely improves, upon currently available datasets would strengthen this manuscript.

The paper was originally conceived primarily as a "data paper" to emphasize the new dataset that focuses on seasonal snow over HMA. It was submitted to Earth System Science Data (ESSD) where we were told it was out of scope because it had "too much analysis" due to the inclusion of analysis of the space-time climatology of seasonal snow. Admittedly, this places this paper somewhere between a typical data paper and a more typical research article that uses existing datasets. The rationale for not including additional analysis was to maintain this paper as primarily a standalone description of a new estimate of seasonal snow climatology over HMA. Including additional analysis through an intercomparison lens will not only push this paper over the length limits, it will likely require giving short shrift to both this new dataset and the other datasets included in the intercomparison. The intercomparison paper we are currently drafting is easily a standalone paper itself and therefore merging the two will, in our opinion, water down both sets of material. Hence our preference is to keep this paper short and to the point in terms of providing a new estimate of seasonal snow climatology, while pointing the readers to the new dataset where further analysis can be performed. In the revised manuscript we will further flesh out the space-time climatology and variability of the new dataset through additional analysis.

On line 62, you say that most reanalysis datasets dot not assimilate snow observations, but on lines 72 and 74 you mention that JRA-55 assimilates ground snow depth and satellite snow cover and ERA5 uses in situ snow depth and satellite snow cover in the assimilation. Please rephase the sentence on line 62 to indicate that some datasets do assimilate snow related observations.

Thank you for your suggestion. We will rephrase the sentence on line 62 as suggested.

You only process tiles with tile-averaged elevation above 1500 m. Do you have an estimate for how much snow is "missed" with this assumption?

It is difficult to explicitly compute how much snow is missed with this spatial screening in our dataset, but the tile-average threshold was chosen to conservatively capture the vast majority of the seasonal mountain snow over HMA. This choice was made for computational reasons to avoid including a significantly larger number of additional tiles that have negligible snow. We can include more areas (at lower elevations) in future versions of this product.

Could you provide a few additional details on your setup of the SSiB3 model? How many snow layers? Could you provide a few additional details of the Liston snow depletion curve? Readers may not look back at previous publications.

We will give more details on the setup of the SSiB3 model, and provide a few additional details of the Liston snow depletion curve in the revised manuscript.

Since you use observations from Landsat 5, 7, and 8, is there any need to do any sort of correction between the three versions? Is the fSCA calculation/band math the same in each version?
The methodology (i.e. end-member mixing analysis) used across Landsat datasets is the same. However, the retrieval uses the specific bands associated with each sensor (i.e. associated with Landsat 5, 7, and 8 ETM, ETM+, and OLI sensors). Intercomparison of fSCA showed no large systematic differences across sensors. It is therefore assumed that any differences that do occur are within the specified Landsat measurement error standard deviation of the products (i.e. 10% of fSCA) that is used in the reanalysis to represent retrieval error/uncertainty.

If additional analysis does make the manuscript too long, I recommend condensing the text on lines 307 to 329 since it states what is already shown in Table 1.
We will condense the text between 307 and 329 as suggested.

---

## Author Response (AR1)

We would like to thank the reviewers for the very helpful and constructive comments in suggesting improvements in our original manuscript. Below we provide point-by-point responses to the comments. Any references to lines of text in the responses below refer to the revised manuscript.

**Reviewer # 1:**
**Major comments:**
As mentioned in my summary above, I was surprised how little additional analysis there was in this manuscript beyond exploring the dataset. Especially since the algorithm development and discussion is published elsewhere, I would like to see additional analysis here. On line 249, you say "A more comprehensive analysis of HMA SWE between multiple products will be addressed in an upcoming intercomparison paper using HMASR." Can that intercomparison not be included here? This did not seem like a very long manuscript, and understanding how the HMASR dataset compared, and likely improves, upon currently available datasets would strengthen this manuscript.

The paper was originally conceived primarily as a "data paper" to emphasize the new dataset that focuses on seasonal snow over HMA. It was submitted to Earth System Science Data (ESSD) where we were told it was out of scope because it had "too much analysis" due to the inclusion of analysis of the space-time climatology of seasonal snow. Admittedly, this places this paper somewhere between a typical data paper and a more typical research article that uses existing datasets. The rationale for not including additional analysis was to maintain this paper as primarily a standalone description of a new estimate of seasonal snow climatology over HMA. Including additional analysis through an intercomparison lens will not only push this paper over the length limits, it will likely require giving short shrift to both this new dataset and the other datasets included in the intercomparison. The intercomparison paper we are currently drafting is easily a standalone paper itself and therefore merging the two will, in our opinion, water down both sets of material. Hence our preference is to keep this paper short and to the point in terms of providing a new estimate of seasonal snow climatology, while pointing the readers to the new dataset where further analysis can be performed. In the revised manuscript we have further fleshed out the space-time climatology and variability of the new dataset through additional analysis, with details listed as below:

1) The 'Results and discussion' section (Sect. 3) is reorganized into three main sub-sections, focusing on the spatial, temporal, and elevational analysis of seasonal SWE.
2) The spatial analysis focuses on climatological peak SWE, peak SWE timing, and the seasonal evolution of SWE, containing the maps (Figure 3-5) and relevant analysis in the previous version.
3) The temporal analysis (Sect. 3.2) is presented in terms of seasonal integrated climatology and interannual variability. Additional analysis is performed and presented in Sect. 3.2.2, examining the variations of seasonal SWE across HMA-wide and at basins scales in different Water Years (with newly added Figure 8 and Figure 9).
4) The elevational analysis (Sect. 3.3) is presented in terms of climatology and variations under different climate conditions. In this section, we have updated the results relevant to peak SWE or peak SWE volumes, using 'pixel-wise peak' SWE instead of 'peak of the annual time-series'. This aims to obtain the maximum snow storage for all elevations, instead of obtaining SWE on the day when snow storage reaches annual maximum. We carefully compared the results and found such changes do not have significant impact on the overall distribution of peak SWE against elevation. Moreover, the variation under different climate conditions is part of the additional analysis we have added in the revised

manuscript, which identifies whether different climate conditions affect the overall snow storage distribution across different elevations. More details can be seen in Sect 3.3.2, with the newly added Table 2, Figure 11, and Figure 12.

5) Minor updates in figures:

    a. shades of +- 1 standard deviation around the climatological mean are added in Figure 6 (according to the editor's suggestion);

    b. line color for Tarim basin is updated to distinguish from that for Yellow and is reflected in Figure 6-7 and Figure 10.

Together, we would argue the set of analyses provides new insight into the space-time climatology of HMA seasonal SWE worthy of a standalone paper.

On line 62, you say that most reanalysis datasets dot not assimilate snow observations, but on lines 72 and 74 you mention that JRA-55 assimilates ground snow depth and satellite snow cover and ERA5 uses in situ snow depth and satellite snow cover in the assimilation. Please rephase the sentence on line 62 to indicate that some datasets do assimilate snow related observations.

Thank you for your suggestion. We have rephrased the sentence on line 62 as suggested. The revised text is updated in line 68 to 69.

You only process tiles with tile-averaged elevation above 1500 m. Do you have an estimate for how much snow is "missed" with this assumption?

It is difficult to explicitly compute how much snow is missed with this spatial screening in the dataset (without performing the reanalysis over those tiles), but the tile-average threshold was chosen a priori to conservatively capture the vast majority of the seasonal mountain snow over HMA. This choice was made for computational reasons to avoid including a significantly larger number of additional tiles that have negligible snow. We will most likely include more areas (at lower elevations) in future versions of this product. This issue has been clarified in line 123 to 126.

Could you provide a few additional details on your setup of the SSiB3 model? How many snow layers? Could you provide a few additional details of the Liston snow depletion curve? Readers may not look back at previous publications.

We have provided more details on the setup of the SSiB3 model, and a few additional details of the Liston snow depletion curve in the revised manuscript. This can be found in line 146 to 151.

Since you use observations from Landsat 5, 7, and 8, is there any need to do any sort of correction between the three versions? Is the fSCA calculation/band math the same in each version?

The methodology (i.e. end-member mixing analysis) used across Landsat datasets is the same. However, the retrieval uses the specific bands associated with each sensor (i.e. associated with Landsat 5, 7, and 8 ETM, ETM+, and OLI sensors). Intercomparison of fSCA showed no large systematic differences across sensors. It is therefore assumed that any differences that do occur are within the specified Landsat measurement error standard deviation of the products (i.e. 10% of fSCA) that is used in the reanalysis to represent retrieval error/uncertainty.

We have clarified this issue in Sect. 2.3.2.

If additional analysis does make the manuscript too long, I recommend condensing the text on lines 307 to 329 since it states what is already shown in Table 1.

We have condense the text between 307 and 329 as suggested. The updated text can be found in section 3.2.1.

**Minor comments:**

On line 174, please write out CDF since it's the first time it appears in the manuscript.
Thank you for the suggestion. We have written out the definition of CDF (in line 212).

In Figure 2, do any tiles have an 18-year average of 0? If so, could you update the colormap/figures to distinguish 0 from non-zero values?
Thank you for the suggestion. We have updated Figure 2 to distinguish 0 from non-zero values.

When describing Figure 3 in the text, could you include the percentage of the domain that is non-seasonal snow/ice?
Thank you for the suggestion. We have included the percentage of the domain that is non-seasonal snow/ice in the text (line 264).

Please rephrase the sentence that begins on line 256 "The median date of peak…". To me, it reads awkwardly.
Thank you for the suggestion. We have revised the text in line 305 to 306.

In Figure 7, the time series for the Ganges-Brahmaputra does not go below ~2 km3. Should those areas that keep snow all year be included in the non-seasonal snow mask?
Thank you for the suggestion. This issue has been clarified in line 377 to 381.

On Figures 3-5, consider including labels of the mountain ranges as you do in Figure 1. Probably not necessary, but I did find myself flipping back to Figure 1 to see the labels since I am not familiar with the ranges in this region.
Thank you for the suggestion. We experimented with Figure 3 and Figure 5 with included mountain labels (shown as below). We found the annotation text a bit distracting when trying to read the maps, and may potentially degrade the quality and presentation of the map. Therefore, we have a slight preference to preserve the original maps.

[Figure]

**Figure 3 (a): Map of pixel-wise peak seasonal SWE climatology, with non-seasonal snow and ice pixels masked out (grey). (b): Map of pixel-wise peak seasonal SWE climatology, without masking of non-seasonal snow and ice pixels for reference.**

[Figure]

**Figure 4: Map of pixel-wise peak seasonal SWE DOWY climatology, with non-seasonal snow and ice pixels masked out (grey). The inset figure is the histogram of peak SWE DOWY. The three dates labeled in the colorbar (DOWY 133, DOWY 169 and DOWY 217) correspond to the 10th, 50th and 90th percentile in the DOWY distribution, and are marked with vertical dashed lines in the inset histogram.**

**Reviewer # 2:**
**Major comments:**
1) I understand the need for a non-seasonal snow mask (based on the remotely-sensed snow cover constraint) but only examining the unmasked areas limits the utility of the analysis and makes the results difficult to compare with other studies. The authors should consider using the modeled melt instead of peak SWE, which should be valid over all the pixels, for the analysis presented in the results.

The reanalysis method will generally work best for seasonal snow where there is a strong signature between snow disappearance and measured fSCA. Therefore, we provide the caveat that non-seasonal snow pixels are likely significantly more erroneous than the seasonal snow pixels. The use of a non-seasonal vs. seasonal snow mask is used in this paper to highlight the part of snow storage that is deemed seasonal snow. In the raw dataset, all pixels are provided and so users are free to take advantage of the non-seasonal snow estimates (with the caveat mentioned above), but for the purposes of highlighting a new estimate of seasonal snow climatology in this paper we prefer to focus on seasonal snow alone. This has been clarified in the revised manuscript (conclusions, line 566 to 571).

2) The authors acknowledge that missing snow cover observations due to clouds will cause higher uncertainty, but do not acknowledge the errors of omission and commission in cloud snow discrimination. These errors will lead to snow that disappears too early or that melts out too late. I'd like to see some discussion of how these errors propagate and are addressed.

More discussion and references to previous work on the methodology and other sources are provided on the impact of clouds and fSCA measurement in the revised manuscript. In summary, the method uses a conservative cloud screening (as outlined in Margulis et al., 2019 and described in more detail below) to limit inclusion of cloudy scenes. This does not prevent errors of omission/commission, but is meant to limit them by mostly including what are most likely high-quality/clear-sky images. Moreover, the reanalysis, unlike other deterministic methods, specifies measurement error in the fSCA time series. This also buffers against direct propagation of fSCA errors into the SWE estimates. More details on this are provided below in response to other comments, and clarified in the revised manuscript (Sect. 2.3.2).

3) Analysis of a spatial timeseries of the datasets show videos of the SWE as being unbelievably smooth and therefore not representing ephemeral snow accurately.

It is acknowledged that the reanalysis method is best designed for non-ephemeral snow where there is a strong seasonal cycle and signal between snow disappearance and measured fSCA that can be captured at the frequency of the fSCA measurements. Hence it is not surprising that ephemeral snow is not well captured. That said, the posterior estimates from the reanalysis tend to be much less smooth than the prior estimates via the incorporation of spatial information contained in the fSCA measurements. Forward modeling estimates (i.e. like that of the prior) tend to be much smoother than those that incorporate a remotely sensed constraint as done here. This has been clarified in the revised manuscript (conclusions, line 572 to 574).

4) Some of the snow albedos are way too low (e.g., 0.01).

This is a result of daily averaging of snow albedo in generating the output files where the original hourly no-snow albedo was stored as zeros. The (modified BATS) snow albedo model used in the reanalysis limits snow albedo to realistic values between ~0.4-0.95. However on days where snow

disappears/appears within the day there will be a mix of zero-valued (i.e. no snow albedo) and sensible snow albedo values, that when averaged can lead to what appear to be values that are "too low". Hence, those days with snow albedo values below ~0.4 should likely be ignored in any analysis. This will be clarified in the data documentation.

5) As the other reviewer notes, in its current form this is a data paper but the submission is listed as a "Research article." Perhaps a journal such as Earth System Science Data would be more appropriate for publication.

Please see comment in response to Reviewer #1, repeated here:

The paper was originally conceived primarily as a "data paper" to emphasize the new dataset that focuses on seasonal snow over HMA. It was submitted to Earth System Science Data (ESSD) where we were told it was out of scope because it had "too much analysis" due to the inclusion of analysis of the space-time climatology of seasonal snow. Admittedly, this places this paper somewhere between a typical data paper and a more typical research article that uses existing datasets. The rationale for not including additional analysis was to maintain this paper as primarily a standalone description of a new estimate of seasonal snow climatology over HMA. Including additional analysis through an intercomparison lens will not only push this paper over the length limits, it will likely require giving short shrift to both this new dataset and the other datasets included in the intercomparison. The intercomparison paper we are currently drafting is easily a standalone paper itself and therefore merging the two will, in our opinion, water down both sets of material. Hence our preference is to keep this paper short and to the point in terms of providing a new estimate of seasonal snow climatology, while pointing the readers to the new dataset where further analysis can be performed. In the revised manuscript we have further fleshed out the space-time climatology and variability of the new dataset through additional analysis, with details listed as below:

6) The 'Results and discussion' section (Sect. 3) is reorganized into three main sub-sections, focusing on the spatial, temporal, and elevational analysis of seasonal SWE.

7) The spatial analysis focuses on climatological peak SWE, peak SWE timing, and the seasonal evolution of SWE, containing the maps (Figure 3-5) and relevant analysis in the previous version.

8) The temporal analysis (Sect. 3.2) is presented in terms of seasonal integrated climatology and interannual variability. Additional analysis is performed and presented in Sect. 3.2.2, examining the variations of seasonal SWE across HMA-wide and at basins scales in different Water Years (with newly added Figure 8 and Figure 9).

9) The elevational analysis (Sect. 3.3) is presented in terms of climatology and variations under different climate conditions. In this section, we have updated the results relevant to peak SWE or peak SWE volumes, using 'pixel-wise peak' SWE instead of 'peak of the annual time-series'. This aims to obtain the maximum snow storage for all elevations, instead of obtaining SWE on the day when snow storage reaches annual maximum. We carefully compared the results and found such changes do not have significant impact on the overall distribution of peak SWE against elevation. Moreover, the variation under different climate conditions is part of the additional analysis we have added in the revised manuscript, which identifies whether different climate conditions affect the overall snow storage distribution across different elevations. More details can be seen in Sect 3.3.2, with the newly added Table 2, Figure 11, and Figure 12.

10) Minor updates in figures:

a. shades of +- 1 standard deviation around the climatological mean are added in Figure 6 (according to the editor's suggestion);
    b. line color for Tarim basin is updated to distinguish from that for Yellow and is reflected in Figure 6-7 and Figure 10.

Together, we would argue the set of analyses provides new insight into the space-time climatology of HMA seasonal SWE worthy of a standalone paper.

**Minor comments:**

**Line 39:**
Please be more descriptive here. What's a localized scale? What's coarse scale vs. fine scale? You've missed all of our papers that focus on SWE over large basins in HIMAT:

Bair, E.H., Stillinger, T., Rittger, K., & Skiles, M. (2021). COVID-19 lockdowns show reduced pollution on snow and ice in the Indus River Basin. Proceedings of the National Academy of Sciences, 118, e2101174118. doi:10.1073/pnas.2101174118

Bair, E.H., Rittger, K., Ahmad, J. A., and Chabot, D. (2020): Comparison of modeled snow properties in Afghanistan, Pakistan, and Tajikistan, The Cryosphere, 14, 331-347, doi: 10.5194/tc-14-331-2020.

Bair, E. H., A. Abreu Calfa, K. Rittger, and J. Dozier (2018), Using machine learning for real-time estimates of snow water equivalent in the watersheds of Afghanistan, The Cryosphere, 12(5), 1579-1594, doi: 10.5194/tc-12-1579-2018.

As there is no universal standard for defining a 'fine scale' and 'coarse scale', we simply classify resolutions around or below 1 km as fine scale, and above 1 km as coarse scale. Similarly, for localized studies we mainly referred to research focusing on basins (including individual small to moderate sized basins), and for regional studies we mainly referred to research on the entire HMA. We provide more description in the revised manuscript (line 43 to 45).

The papers that you listed above are great works in the HMA domain, and we have include all of them in our literature review (line 40; line 77 to 79).

**Line 117:**
That cutoff is too high in northern HMA. For example, central Almaty KZ (el 800m) has 0.5 m of snow on the ground in January & February (http://www.pogodaiklimat.ru/climate/36870.htm). Could you explain further how the cutoff value was selected, and moreover why a cutoff is needed?

The cutoff using a tile-average elevation above 1500 m was mainly chosen as a constraint on computational cost. When embarking on this study, a probabilistic snow reanalysis at this resolution/extent had not been created and computational compromises were made due to the large computational cost. It was an efficient threshold for most areas of HMA that avoided running the reanalysis at tiles with little to no seasonal snow. We acknowledge that this threshold might exclude snow is some areas of the domain and have clarify this in the revised manuscript (line 123 to 126). We anticipate that this threshold will be relaxed or removed in future versions of this product.

**Line 147:**
Because of the monsoon, HIMAT is much cloudier than the Sierra Nevada or Andes. This is an obstacle for optical sensors.

We agree that the HMA region is much cloudier than the Sierra Nevada or Andes, where optical sensors may not provide as much information and are subject to additional errors. We have pointed out this limitation in the revised manuscript (line 171-173) and have provided more details when mentioned in the text.

**Line 156:**
This is a very useful study that I had not seen before, however I'm skeptical about the accuracy of the remotely-sensed snow cover ablation retrievals, especially in the monsoon-dominated parts of HIMAT. Snow cloud discrimination remains an unsolved problem (Stillinger et al 2019) that plagues MODSCAG and every other snow cover product that relies on optical sensors.

Given that the keyword "clouds" is not even mentioned in Liu and Margulis (2019), I can only assume that there are many times when clouds are mistaken for snow and vice-versa, leading to erroneous fSCA conditioning where the snow cover melts out earlier than reality or persists later.

Stillinger, T., Roberts, D. A., Collar, N. M., & Dozier, J. (2019). Cloud Masking for Landsat 8 and MODIS Terra Over Snow-Covered Terrain: Error Analysis and Spectral Similarity Between Snow and Cloud. Water Resources Research, 55, 6169-6184. https://doi.org/10.1029/2019wr024932

Here are our responses:
1) For the comment 'Given that the keyword "clouds" is not even mentioned in Liu and Margulis (2019) …', we would like to first clarify that Liu and Margulis (2019) and Margulis et al. (2019) are two companion papers. Liu and Margulis (2019) is more focused on methods of snowfall parameterization, while details of fSCA processing were given in Margulis et al. (2019). Cloud screening has been performed in both Landsat and MODSCAG retrievals, as described in Margulis et al. (2019). Specifically, for Landsat, any tile with a diagnosed cloud cover fraction of greater than 40% is excluded entirely. For MODSCAG, only "near-nadir" tiles are included and, of those, any tile with a diagnosed cloud cover fraction of greater than 10% is excluded entirely. This subset of Landsat and MODSCAG tiles for inclusion therefore uses a conservative screening meant to exclude cloudy tiles. This does not prevent errors of omission/commission, but is meant to mitigate cloud impacts by not including all tiles. More detail on this has been provided in the revised manuscript (Sect. 2.3.2).
2) For the comment 'I'm skeptical about the accuracy of the remotely-sensed snow cover ablation retrievals, especially in the monsoon-dominated parts of HIMAT…', we acknowledge that the accuracy of fSCA retrievals are likely not as good (or at least more subject to omission/commission errors due to clouds) in the monsoon dominated parts of HMA, which in our case excludes many more Landsat/MODSCAG measurements.
3) For the comment 'Snow cloud discrimination remains an unsolved problem (Stillinger et al 2019) that plagues MODSCAG and every other snow cover product that relies on optical sensors.' and 'I can only assume that there are many times when clouds are mistaken for snow and vice-versa, leading to erroneous fSCA conditioning where the snow cover melts out earlier than reality or persists later'. The general point raised regarding cloud

classification problems with optical sensors is a good one and one we highlight in more detail in the revised manuscript. It is true that any study using estimates derived from optical sensors will be subject to some level of these errors. But here lies an important point regarding the reanalysis methodology used in this manuscript vs. other methods. In deterministic "SWE reconstruction" methods (i.e. Bair et al., 2020), the fSCA measurements are used directly to estimate ablation rates, i.e., the ablation rate is effectively obtained by interpolating between measurements. This is equivalent to assuming no measurement error in fSCA (despite the known errors cited above). Such a method will directly propagate errors (including and especially those of omission/commission) to the SWE estimates. In contrast, the SWE reanalysis method used here explicitly acknowledges measurement error in the fSCA measurements used. So rather than interpolating between fSCA measurements, a reanalysis (data assimilation) approach is more akin to a least-squares fit of the data, i.e., one that acknowledges error and does not "overfit" in an interpolation sense. Hence propagation of error is reduced. The inclusion of measurement error in the form used in this manuscript (i.e., constant 10% and 15% error standard deviations for Landsat and MODSCAG respectively) is undoubtedly a simplification of the real error scenario, but the method at least allows for its acknowledgement and representation in a bulk sense. Future versions could even attempt to include a more refined representation of these and other error types. The combination of a conservative cloud screening process (described above) and the ability to account for bulk measurement errors in the reanalysis methodology provides, in our opinion, a "best-case" mitigation of inherent cloud-based errors. More detail on has been provided in the revised manuscript (Sect. 2.3.2).

References:
Margulis, S. A., Girotto, M., Cortés, G. and Durand, M.: A Particle Batch Smoother Approach to Snow Water Equivalent Estimation, Journal of Hydrometeorology, 16(4), 1752–1772, 2015.
Margulis, S. A., Liu, Y. and Baldo, E.: A Joint Landsat- and MODIS-Based Reanalysis Approach for Midlatitude Montane Seasonal Snow Characterization, Front. Earth Sci., 7, 4257, doi:10.3389/feart.2019.00272, 2019.
Liu, Y. and Margulis, S. A.: Deriving Bias and Uncertainty in MERRA-2 Snowfall Precipitation Over High Mountain Asia, Front. Earth Sci., 7, 39, doi:10.3389/feart.2019.00280, 2019.

**Line 173:**
According to Margulis et al (2019), the cloud screening relies on CFMask (Landsat) and the MOD09GA QA bits (MODIS). These masks have a precision of 0.70 & 0.17 and a recall of 0.86 & 0.72 (Stillinger et al. 2019). Thus, again this calls into question the accuracy of the remotely-sensed fSCA.
See responses to comments above. More detail on this are provided in the revised manuscript (Sect. 2.3.2).

**Line 180:**
The authors need to acknowledge that the cloudy images create an issue not only of missing observations, but of false positives (low precision for cloud masks) and false negatives (low recall

for cloud masks) for snow detection, which in turn will create snow ablation curves that do not represent that snow cover on the ground.

See responses to comments above. Specifically, snow ablation curves are not explicitly created from the fSCA time series in the way they are for SWE reconstruction methods. Hence, as described above, these errors are mitigated using the proposed approach. More detail on this are provided in the revised manuscript (Sect. 2.3.2).

**Line 199:**

RGI is more appropriate here, as its a snapshot of glaciers around 2000 whereas GLIMS (which includes RGI) contains outlines from a much larger date range.

This has been clarified in the revised manuscript (line 240 to 241).

**Line 221:**

Can you elaborate on how the Bayesian update process was performed on these pixels? In the User Guide at NSIDC, the authors state that these pixels are given a type 1 designation; that is having no prior simulation. A further description is warranted here.

Actually, there is no difference in the Bayesian update between 'seasonal snow pixels' and 'non-seasonal snow pixels'. Estimates at both types of pixels were computed in the same way and included in the HMASR dataset. It is mainly an external mask of 'non-seasonal snow and ice' that we provide to the users (and ourselves), given the fact that we think the results are less accurate compared to other seasonal snow estimates. Moreover, type 1 designation was not assigned to 'seasonal snow pixels' or 'non-seasonal snow pixels' in our NSIDC documentation. Some pixels have little snowfall based on our prior simulation, and we skipped running those pixels in the reanalysis (to reduce computational expense), and assigned them as type 1. This has been clarified in the revised manuscript (conclusions, line 566 to 571).

**Line 228:**

That is in agreement with measurements and SWE reconstructions from Salang Pass in Afghanistan (35N, 69E, el 3366m), one of the few high alititude sites with snow climate records in Afghanistan (Bair et al 2018)

Bair, E. H., Abreu Calfa, A., Rittger, K., & Dozier, J. (2018). Using machine learning for real-time estimates of snow water equivalent in the watersheds of Afghanistan. The Cryosphere, 12, 1579-1594. https://doi.org/10.5194/tc-12-1579-2018

It is good to know that they are in agreement. We have added this reference to the revised manuscript (line 275 to 276).

**Line 240:**

and errors of omission and commission in cloud/snow identification

This has been included in revised manuscript (Sect. 2.3.2).

**Line 300:**

Also most studies don't mask out non-seasonal SWE. For these areas with perennial snow/ice, the computed melt should still be valid, and is likely useful for water managers. Thus, it would be

useful to instead show the melt volumes including the masked areas, as just showing the seasonal SWE is misleading.

As mentioned above, there is no difference in the Bayesian update between 'seasonal snow pixels' and 'non-seasonal snow pixels'. Both types of pixels were fully computed and results were included the HMASR dataset. It is mainly an external mask of 'non-seasonal snow and ice' that we provide to the users, given the fact that we think the results are less accurate compared to other seasonal snow estimates. This has been clarified in the revised manuscript (conclusions, line 566 to 571).

---

## Author Response (AR2)

We would like to thank the reviewers for the very helpful and constructive comments in suggesting improvements in our original manuscript. Below we provide point-by-point responses to the comments.

**Reviewer # 1:**

I appreciate the additional analysis and the thorough response from the authors. I feel the authors have adequately addressed my comments, and I think they also had good responses to Reviewer #2's concerns. Therefore, I think this manuscript should be accepted for publication and I look forward to reading the forthcoming intercomparison manuscript. I only have one minor comment below:

For Figure 9, assuming I'm understanding the red line correctly (an average of the white 'x's?), could you double check your calculations? For the Indus and Salween, for example, it appears that most of the 'x's/date of basin-scale peak SWE volume occur after the red line. I would have assumed it would be shifted closer to the 'x' marks? Perhaps I'm misunderstanding this.

Thank you for raising that point. We have carefully checked Figure 9, and found the calculations are correct for the date of peak SWE volume occurrences within each year (the white 'x's) and the climatology (red line). However, we noticed that the location of these white 'x's are offset towards the right, as an result of error in the plotting code. This fully explains why 'most of the 'x's/date of basin-scale peak SWE volume occur after the red line' as pointed out in your comment. We have fixed the error and the corrected figures are shown below (also updated in the manuscript):

[Figure]

**Figure 8: Daily time-series of HMA-wide SWE volumes displayed as functions of DOWY and WY. The symbol 'x' is used to mark the date of peak SWE volume occurrence, with the corresponding peak SWE volume labeled in each WY. The symbol 'o' is used to mark the dates when 50% of the peak SWE volume is reached in each WY. The vertical red line is used to indicate the 18-year average timing of HMA-wide peak SWE volume.**

[Figure]

**Figure 9: Daily time-series of basin-scale SWE volumes displayed as functions of DOWY and WY. The symbol 'x' is used to mark the date of basin-scale peak SWE volume occurrence in each WY. The vertical red line is used to indicate the 18-year average timing of basin-scale peak SWE volume.**